# Cell cycle-specific loading of condensin I is regulated by the N-terminal tail of its kleisin subunit

Shoji Tane[1], Keishi Shintomi[1], Kazuhisa Kinoshita[1], Yuko Tsubota[2], Makoto M Yoshida[1], Tomoko Nishiyama[2], Tatsuya Hirano[1]*

[1]Chromosome Dynamics Laboratory, RIKEN, Wako, Japan; [2]Division of Biological Sciences, Graduate School of Science, Nagoya University, Nagoya, Japan

**Abstract** Condensin I is a pentameric protein complex that plays an essential role in mitotic chromosome assembly in eukaryotic cells. Although it has been shown that condensin I loading is mitosis specific, it remains poorly understood how the robust cell cycle regulation of condensin I is achieved. Here, we set up a panel of in vitro assays to demonstrate that cell cycle-specific loading of condensin I is regulated by the N-terminal tail (N-tail) of its kleisin subunit CAP-H. Deletion of the N-tail accelerates condensin I loading and chromosome assembly in *Xenopus* egg mitotic extracts. Phosphorylation-deficient and phosphorylation-mimetic mutations in the CAP-H N-tail decelerate and accelerate condensin I loading, respectively. Remarkably, deletion of the N-tail enables condensin I to assemble mitotic chromosome-like structures even in interphase extracts. Together with other extract-free functional assays in vitro, our results uncover one of the multilayered mechanisms that ensure cell cycle-specific loading of condensin I onto chromosomes.

## Editor's evaluation

This important study provides insight into how vertebrate condensin I activity is restricted to mitosis. Using in vitro experiments in *Xenopus* extracts, convincing evidence is presented that phosphorylation of the N-terminal extension of CAP-H relieves an inhibitory activity that prevents condensin I loading onto chromosomes. The authors present a speculative model for the mechanistic basis of this inhibition which will provide inspiration for future investigations in the chromosome condensation field.

*For correspondence:
hiranot@riken.jp

**Competing interest:** The authors declare that no competing interests exist.

## Introduction

Chromosome assembly is an essential cellular process that ensures equal segregation of genetic information into two daughter cells during mitosis (*Batty and Gerlich, 2019*; *Paulson et al., 2021*). Extensive studies during the past two decades or so have established the consensus that the condensin complexes play a central role in this process (*Hirano, 2016*; *Uhlmann, 2016*). Many eukaryotic species have two different condensin complexes, known as condensins I and II, although some species including fungi have only condensin I. The two condensin complexes, each of which is composed of five subunits, share a pair of SMC (structural maintenance of chromosome) ATPase subunits (SMC2 and SMC4) but are distinguished by distinct sets of non-SMC subunits (*Ono et al., 2003*). In condensin I, the kleisin subunit CAP-H bridges the head domains of a V-shaped SMC2–SMC4 heterodimer, and is then bound by a pair of HEAT subunits, CAP-D2 and -G (*Figure 1A*, right). In vertebrates, the two condensin complexes cooperate to assemble rod-shaped chromosomes in mitosis (*Ono et al., 2003*; *Shintomi and Hirano, 2011*; *Green et al., 2012*; *Gibcus et al., 2018*), but they display differential

subcellular localization during the cell cycle. In HeLa cells, for instance, condensin I primarily resides in the cytoplasm during interphase, and gets loaded onto chromosomes immediately after the nuclear envelope breaks down in prometaphase (*Ono et al., 2004*; *Hirota et al., 2004*). However, mitosis-specific loading of condensin I can be recapitulated in membrane-free *Xenopus* egg extracts (*Hirano et al., 1997*), suggesting that a mechanism(s) independent of the regulation of subcellular localization also operates to ensure the mitosis-specific action of condensin I. Thus, a whole molecular picture of how the loading and action of condensin I are tightly regulated throughout the cell cycle remains to be determined.

Early studies demonstrated that condensin I is phosphorylated in a mitosis-specific manner in *Xenopus* egg extracts (*Hirano et al., 1997*), and that the positive supercoiling activity of condensin I can be activated by Cdk1 phosphorylation in vitro (*Kimura et al., 1998*; *Kimura et al., 2001*). More recently, a mitotic chromatid reconstitution assay using purified proteins was used to demonstrate that Cdk1 phosphorylation of condensin I allows the complex to load onto chromosomes and to drive chromatid assembly (*Shintomi et al., 2015*). The major targets of mitotic phosphorylation identified in these studies included the CAP-D2 and CAP-H subunits. Phosphorylation of condensin I by other mitotic kinases has also been reported (*Lipp et al., 2007*; *Takemoto et al., 2007*; *St-Pierre et al., 2009*). All subunits of condensin I are large polypeptides of >100 kD, making it a daunting challenge to identify all phosphorylation sites and dissect their functional impacts. A global proteomic analysis of Cdk1 phosphorylation sites provided a clue to potentially relieving this problem, however (*Holt et al., 2009*). This study showed that Cdk1 phosphorylation often occurs in intrinsically disordered regions (IDRs) of proteins whose evolutionary conservation is relatively poor. It seems that this rule can be applied to the subunits of condensin I as well (*Bazile et al., 2010*).

In the current study, we focus on the N-terminal IDR of the vertebrate kleisin subunit CAP-H. We show that this region, which is referred to as the CAP-H N-tail, acts as a negative regulatory element for condensin I function. Deletion of the N-tail accelerates condensin I loading and mitotic chromosome assembly in *Xenopus* egg extracts. Phosphorylation-deficient mutations in the N-tail decelerate condensin I loading, whereas phosphorylation-mimetic mutations accelerate this process. Remarkably, when the N-tail function is compromised, the resultant mutant forms of condensin I enable the assembly of mitotic chromosome-like structures even in interphase extracts. The N-tail deletion mutant is also characterized in topological loading and loop extrusion assays in vitro. Taken together, our results uncover one of the multilayered mechanisms of cell cycle regulation of condensin I.

## Results

### The N-terminal tail of the kleisin subunit CAP-H

The kleisin subunit CAP-H of condensin I has five sequence motifs widely conserved among eukaryotes (*Figure 1A*; *Hara et al., 2019*; *Kinoshita et al., 2022*). Among them, motifs I and V bind to the SMC2 neck and the SMC4 cap regions, respectively (*Hassler et al., 2019*), whereas motifs II and IV interact with CAP-D2 and CAP-G, respectively (*Piazza et al., 2014*; *Kschonsak et al., 2017*; *Hara et al., 2019*). In the current study, we focus on the N-terminal extension, located upstream of motif I, that is shared by the vertebrate CAP-H orthologs (*Figure 1A*). This extension, the CAP-H N-tail, has several characteristic features as summarized below. Firstly, it is ~80 amino-acid long in vertebrates, but the corresponding extension is much shorter or completely missing in fungi (*Bazile et al., 2010*). Secondly, although most of the N-tail is predicted to be structurally disordered, a stretch of ~17-amino-acid long, located in its middle region, is conserved among vertebrates and is predicted to form an α-helix, as judged by both the conventional secondary structure prediction software Jpred4 (https://www.compbio.dundee.ac.uk/jpred/) and AlphaFold2 (https://alphafold.ebi.ac.uk/). Thirdly, the vertebrate N-tail contains multiple SP/TP sites, sites often targeted by cyclin–CDK complexes, although their number varies among different species.

### Deletion of the CAP-H N-tail accelerates condensin I loading and mitotic chromosome assembly

To address the role of the CAP-H N-tail in condensin I function, we expressed mammalian subunits of condensin I using a baculovirus expression system in insect cells, and purified recombinant complexes according to the procedure described previously (*Kinoshita et al., 2015*; *Kinoshita et al., 2022*). In

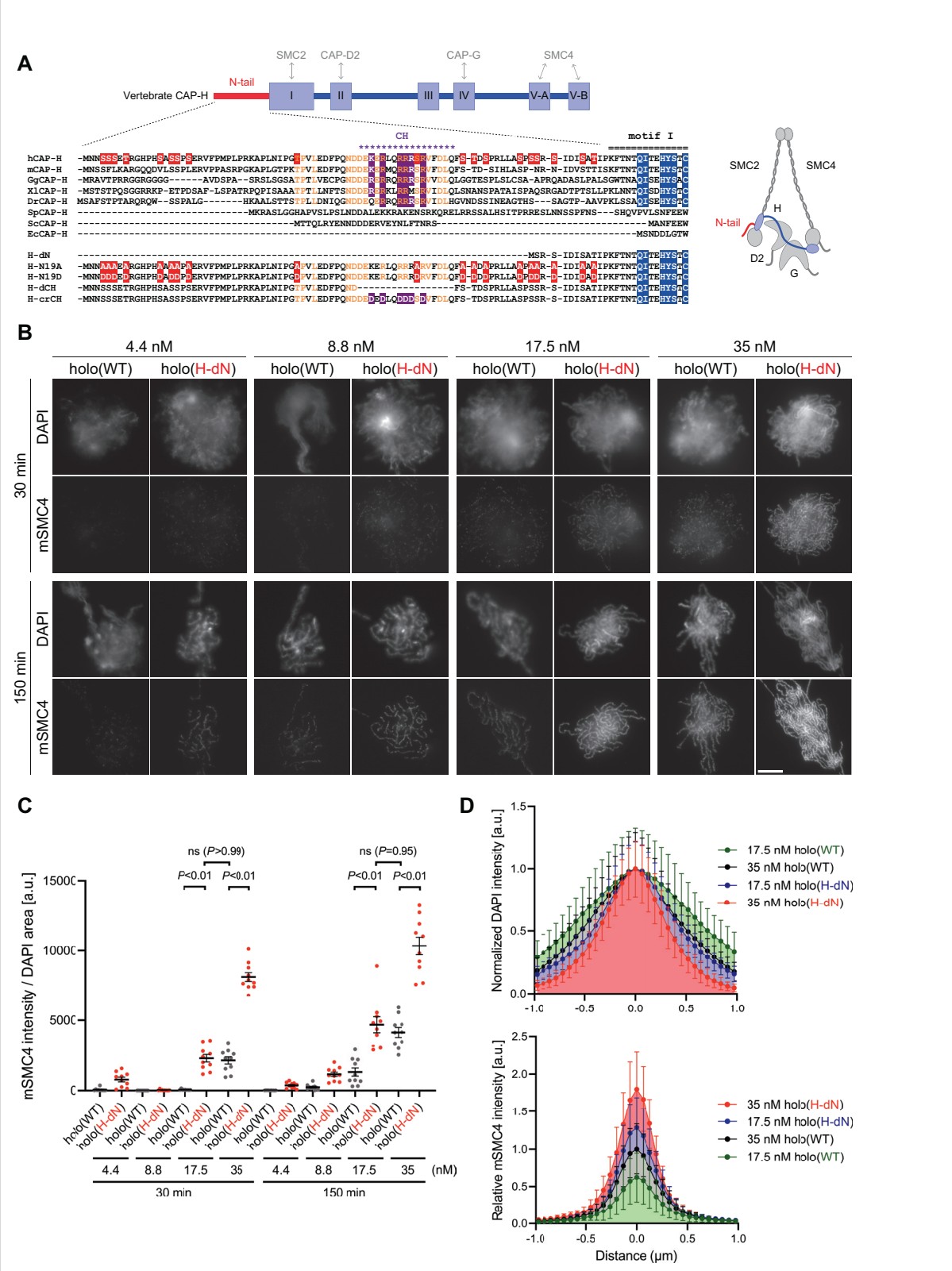

**Figure 1.** Deletion of the CAP-H N-tail accelerates condensin I loading and mitotic chromosome assembly. (**A**) Domain organization of vertebrate CAP-H and sequence alignments of the CAP-H N-tail in eukaryotes (left). Shown in the upper half is an alignment of the CAP-H orthologs (*Homo sapiens*: hCAP-H, *Mus musculus*: mCAP-H, *Gallus gallus*: GgCAP-H, *Xenopus laevis*: XlCAP-H, *Danio rerio*: DrCAP-H, *Schizosaccharomyces pombe*: SpCAP-H, *Saccharomyces cerevisiae*: ScCAP-H, and *Encephalitozoon cuniculi*: EcCAP-H). Six mutants tested in the current study (H-dN, H-N19A,

*Figure 1 continued*

H-N19D, H-dCH, and H-crCH) are shown in the bottom half. Conserved amino-acid residues were shown in yellow (N-tail) or blue (motif I). The helix motif predicted using Jpred4 (http://www.compbio.dundee.ac.uk/jpred/) is indicated by 'CH' (for the conserved helix). The N19A/N19D and crCH mutation sites were shown in red and purple, respectively. Also shown in a schematic diagram of the architecture of vertebrate condensin I (right). (**B**) Add-back assay with holo(WT) and holo(H-dN). Mouse sperm nuclei were incubated with condensin-depleted M-HSS that had been supplemented with holo(WT) or holo(H-dN) at concentrations of 4.4, 8.8, 17.5, and 35 nM. After 30 and 150 min, the reaction mixtures were fixed and processed for immunofluorescence labeling with an antibody against mSMC4. DNA was counterstained with DAPI. Shown here is a representative image from over 10 chromosome clusters examined per condition. Scale bar, 10 µm. (**C**) Quantification of the intensity of mSMC4 per DNA area in the experiment shown in (B) (*n* = 10 clusters of chromosomes). The error bars represent the mean ± standard error of the mean (SEM). The p values were assessed by Tukey's multiple comparison test after obtaining a significant difference with two-way analysis of variance (ANOVA). (**D**) Line profiles of mitotic chromosomes observed at 150 min shown in the experiment shown in (B). Signal intensities of DAPI (top) and mSMC4 (bottom) from chromosomes assembled by holo(WT) or holo(H-dN) at 17.5 or 35 nM were measured along with the lines drawn perpendicular to chromosome axes (*n* = 20). The mean and standard deviation (SD) were normalized individually to the DAPI intensities (arbitrary unit [a.u.]) at the center of chromosome axes (distance = 0 µm) (top). Intensities of mSMC4 signals were normalized relative to the value from holo(WT) at 35 nM (bottom). The error bars represent the mean ± SD.

The online version of this article includes the following source data and figure supplement(s) for figure 1:

**Source data 1.** Microsoft excel of non-normalized data corresponding to *Figure 1C*.

**Source data 2.** Microsoft excel of non-normalized data corresponding to *Figure 1D*.

**Figure supplement 1.** Recombinant condensin I complexes and immunodepletion.

**Figure supplement 1—source data 1.** Raw data uncropped gel corresponding to *Figure 1—figure supplement 1A, B*.

**Figure supplement 1—source data 2.** Microsoft excel of DNA constructs used in this study.

**Figure supplement 1—source data 3.** Raw data uncropped blots corresponding to *Figure 1—figure supplement 1C*.

**Figure supplement 2.** Deletion of and mutations in the conserved helix accelerates condensin I loading and mitotic chromosome assembly.

**Figure supplement 2—source data 1.** Microsoft excel of non-normalized data corresponding to *Figure 1—figure supplement 2B*.

**Figure supplement 2—source data 2.** Microsoft excel of non-normalized data corresponding to *Figure 1—figure supplement 2C*.

the first set of experiments, a holocomplex containing wild-type hCAP-H, holo(WT), and a holocomplex containing mutant hCAP-H that lacks its N-terminal 77 amino acids, holo(H-dN), were prepared (*Figure 1A* and *Figure 1—figure supplement 1A*). We then tested the ability of these recombinant complexes to assemble mitotic chromosomes in *Xenopus* egg extracts (high-speed supernatants of metaphase-arrested extracts, hereafter referred to as M-HSS) that had been immunodepleted of endogenous condensin subunits (*Kinoshita et al., 2015*; *Kinoshita et al., 2022*; *Figure 1—figure supplement 1C*). The condensin-depleted extracts were supplemented with increasing concentrations (4.4, 8.8, 17.5, and 35 nM) of holo(WT) or holo(H-dN), mixed with mouse sperm nuclei and incubated at 22°C. Aliquots were taken from the reaction mixtures at 30 and 150 min, fixed and processed for immunofluorescence using an antibody against the recombinant subunit mSMC4 (*Figure 1B*). We found that holo(WT) produced a cluster of rod-shaped chromosomes with mSMC4-positive axes under the standard condition (i.e., 35-nM condensin I, 150-min incubation) set up in the previous study (*Kinoshita et al., 2022*). As expected, progressively poorer assembly was observed with decreasing concentrations of condensin I (17.5–4.4 nM) or at the earlier time point (30 min). Remarkably, we found that significantly higher levels of holo(H-dN) were detected on chromatin than holo(WT) at both time points and at the different concentrations of condensin I tested (*Figure 1B, C*). It was also noticed that the chromosomes assembled with 35 nM holo(H-dN) at 150 min were much thinner than those assembled with 35 nM holo(WT) at 150 min (*Figure 1B, D*). When the concentration of holo(H-dN) was reduced to 17.5 nM, the resultant chromosomes were comparable with those assembled with 35 nM holo(WT), in terms of both the morphology and the mSMC4 signal levels (*Figure 1C, D*). Taken together, we concluded that the deletion of the CAP-H N-tail accelerates the loading of condensin I on chromosomes in M-HSS, and that the chromosomal levels of condensin I are tightly correlated with the chromosome morphology produced.

To gain further insights into the functional contribution of the CAP-H N-tail to condensin I-mediated chromosome assembly, we next focused on the conserved helix (CH), located in the middle of the N-tail, that contains conserved basic amino acids (*Figure 1A*). We constructed a holocomplex containing hCAP-H that lacks the CH, holo(H-dCH), and a holocomplex containing hCAP-H in which the basic amino acids within the CH are substituted with acidic residues, holo(H-crCH) (*Figure 1A* and *Figure 1—figure supplement 1A*). Here, dCH and crCH stand for deletion of the CH and

charge-reversed CH, respectively. We found that the holo(H-dCH) and holo(H-crCH) exhibited a phenotype very similar to that of holo(H-dN) in terms of both increased condensin I loading and resultant chromosome morphology (*Figure 1—figure supplement 2A–C*). Thus, the CH and its positive charges play an important role in negatively regulating condensin I loading in M-HSS.

## Phosphorylation-deficient and phosphorylation-mimetic mutations of the CAP-H N-tail decelerate and accelerate condensin I loading, respectively

We next wished to understand how phosphorylation of the CAP-H N-tail might affect its function. Previous studies had shown that *Xenopus* CAP-H is phosphorylated in a mitosis-specific manner in *Xenopus* egg extracts (*Hirano et al., 1997*; *Kimura et al., 1998*), and that *Xenopus* and human CAP-H can be phosphorylated by cyclin B-Cdk1 in vitro (*Kimura et al., 1998*; *Kimura et al., 2001*; *Shintomi et al., 2015*). Whereas hCAP-H has three SP sites and one TP site in its N-tail (*Figure 2—figure supplement 1A*), it is known that Cdk1 phosphorylation is not exclusively proline directed (*Brown et al., 2015*; *Suzuki et al., 2015*; *Krasinska and Fisher, 2022*). Moreover, a phospho-proteomic analysis identified phosphorylation at both SP/TP and non-SP/TP sites in this region (*Hornbeck et al., 2015*). Thus, the targets of mitosis-specific phosphorylation in CAP-H have not yet been fully characterized.

To test whether phosphorylation of the hCAP-H N-tail plays an important role in condensin I loading, we prepared a holocomplexes harboring phosphorylation-deficient mutations in the CAP-H N-tail, holo(H-N19A), where all serines and threonines present in the N-tail were substituted with alanines (*Figure 1A* and *Figure 1—figure supplement 1A*). When the holo(WT) and holo(H-N19A) were incubated with M-HSS, two phosphoepitopes (pS17 and pS76) were detectable in the former, but not in the latter (*Figure 2—figure supplement 1A, B*; Materials and methods). Neither of the epitopes was detected in both complexes that had been incubated with interphase HSS (I-HSS), indicating that the corresponding sites are phosphorylated in holo(WT) in a mitosis-specific manner. Holo(H-N19A) was then subjected to the add-back assay in M-HSS. We found that the loading of holo(H-N19A) onto chromatin was greatly reduced compared to holo(WT) or holo(H-dN) (*Figure 2A–C*). The resultant chromosomes produced by holo(H-N19A) were poorly organized and abnormally thick, having hazy surfaces (*Figure 2A–C*).

To further substantiate the observations described above, we then prepared and tested a holocomplex harboring phosphorylation-mimetic mutations, holo(H-N19D) (*Figure 1A* and *Figure 1—figure supplement 1A*), in which all serines and threonines in the N-tail were substituted with aspartic acids. Remarkably, we found that holo(H-N19D) behaved very similarly to holo(H-dN) in the add-back assay (*Figure 2D–F*). Although we cannot rule out the possibility that the introduction of multiple mutations into the N-tail causes unforeseeable adverse effects on protein conformations, these results supported the idea that multisite phosphorylation in the CAP-H N-tail relieves its negative effects on condensin I loading and mitotic chromosome assembly.

## Deletion of the CAP-H N-tail enables condensin I to assemble mitotic chromosome-like structures even in interphase extracts

Having observed the accelerated loading of holo(H-dN) in M-HSS, we wondered how the mutant complex would behave in interphase extracts. To eliminate potential complications from the subcellular localization of condensin I (*Ono et al., 2004*; *Shintomi and Hirano, 2011*), we used membrane-free I-HSS in the following experiments in which no nuclear envelope was assembled around the chromatin introduced. Consistent with the previous study (*Hirano et al., 1997*), endogenous *Xenopus* condensins failed to associate with chromatin in I-HSS, leaving a spherical compact mass, although they bound to chromatin and assembled a cluster of rod-shaped chromosomes in M-HSS (*Figure 3A*). We then monitored the morphological changes of sperm nuclei in a condensin-depleted I-HSS. We found that the nuclei were transiently swollen at early time points (15–30 min) and then converted into compact chromatin masses at a late time point (150 min) (*Figure 3B*). The terminal morphology was very similar to that observed in the undepleted I-HSS described above (*Figure 3A*). The same was true in the condensin-depleted extract that had been supplemented with 35 nM holo(WT). Signals of mSMC4 were hardly observed on the chromatin masses under this condition (*Figure 3B*). When the condensin-depleted extract was supplemented with 35 nM holo(H-dN), however, significant levels of

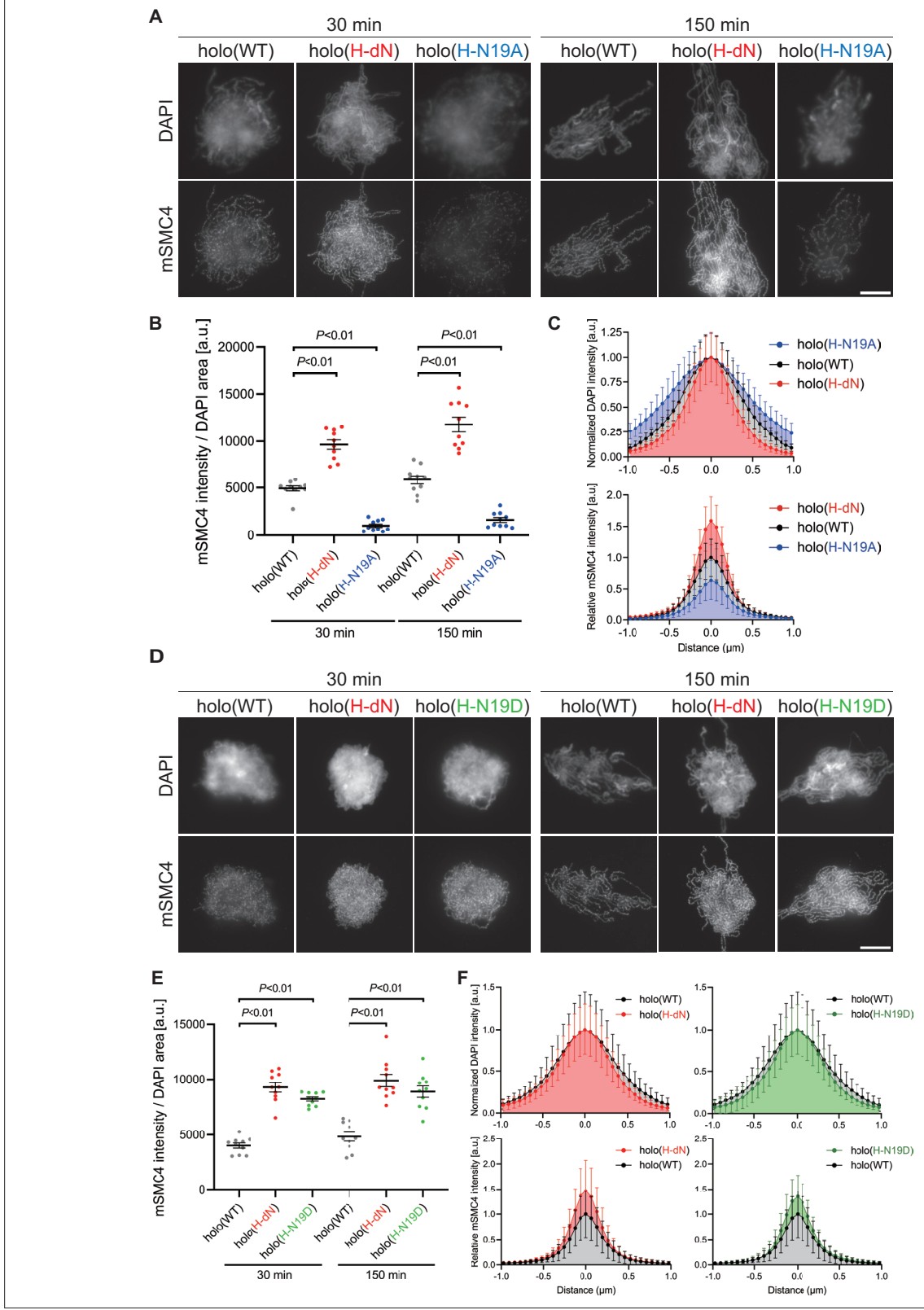

**Figure 2.** Phosphorylation-deficient and phosphorylation-mimetic mutations of the CAP-H N-tail decelerate and accelerate condensin I loading, respectively. (**A**) Mouse sperm nuclei were incubated with condensin-depleted M-HSS that had been supplemented with holo(WT), holo(H-dN), or holo(H-N19A) at a final concentration of 35 nM. After 30 and 150 min, the reaction mixtures were fixed and processed for immunofluorescence labeling with an antibody against mSMC4. DNA was counterstained with DAPI. Shown here is a representative image from over 10 chromosome clusters

*Figure 2 continued on next page*

*Figure 2 continued*

examined per condition. Scale bar, 10 μm. (**B**) Quantification of the intensity of mSMC4 per the DAPI area in the experiment shown in (A) (*n* = 10 clusters of chromosomes). The error bars represent the mean ± standard error of the mean (SEM). The p values were assessed by Tukey's multiple comparison test after obtaining a significant difference with one-way analysis of variance (ANOVA) at each time point. A dataset from a single representative experiment out of more than three repeats is shown. (**C**) Line profiles of mitotic chromosomes observed at 150 min in the experiment shown in (A). Signal intensities of DAPI (top) and mSMC4 (bottom) of the chromosomes assembled with holo(WT) (black), holo(H-dN) (red), or holo(H-N19A) (blue) were measured along with the lines drawn perpendicular to chromosome axes (*n* = 20). The mean and standard deviation were normalized individually to the DAPI intensities (arbitrary unit [a.u.]) at the center of chromosome axes (distance = 0 μm) within each set. Intensities of mSMC4 signals from holo(H-dN) and holo(H-N19A) were normalized relative to the value from holo(WT). A dataset from a single representative experiment out of more than three repeats is shown. (**D**) Add-back assay using holo(WT), holo(H-dN), and holo(H-N19D) at a final concentration of 35 nM was performed as described in (A). Shown here is a representative image from over 10 chromosome clusters examined per condition. Scale bar, 10 μm. (**E**) Quantification of the intensity of mSMC4 per the DAPI area in the experiment shown in (D) (*n* = 10 clusters of chromosomes). The error bars represent the mean ± SEM. The p values were assessed by Tukey's multiple comparison test after obtaining a significant difference with one-way ANOVA at each time point. A dataset from a single representative experiment out of more than three repeats for holo(WT) and holo(H-dN) and two repeats for holo(H-N19D) is shown. (**F**) Line profiles of mitotic chromosomes assembled observed at 150 min in the experiment shown in (D). The signal intensities of DAPI (top) and mSMC4 (bottom) of the chromosomes assembled with holo(WT) (black), holo(H-dN) (red), or holo(H-N19D) (green) were measured and plotted as described in (C) (*n* = 20). A dataset from a single representative experiment out of more than three repeats for holo(WT) and holo(H-dN) and two repeats for holo(H-N19D) is shown.

The online version of this article includes the following source data and figure supplement(s) for figure 2:

**Source data 1.** Microsoft excel of non-normalized data corresponding to *Figure 2B*.

**Source data 2.** Microsoft excel of non-normalized data corresponding to *Figure 2C*.

**Source data 3.** Microsoft excel of non-normalized data corresponding to *Figure 2E*.

**Source data 4.** Microsoft excel of non-normalized data corresponding to *Figure 2F*.

**Figure supplement 1.** Mitosis-specific phosphorylation of condensinI subunits.

**Figure supplement 1—source data 1.** Raw data uncropped blots corresponding to *Figure 2—figure supplement 1B*.

**Figure supplement 1—source data 2.** Raw data uncropped blots corresponding to *Figure 2—figure supplement 1C*.

**Figure supplement 1—source data 3.** Raw data uncropped blots corresponding to *Figure 2—figure supplement 1D*.

mSMC4 signals were detectable on chromatin, especially at early time points (*Figure 3B*). It was also noticed that the compact masses observed at 150 min had somewhat irregular surfaces.

To further extend the observations obtained at the standard concentration (35 nM) of condensin I added back into the extracts, we tested a higher concentration (150 nM) of holo(WT) and holo(H-dN) in the same assay. We found that 150 nM holo(WT) produced a temporal series of chromatin masses that was very similar to that observed with 35 nM holo(H-dN). In contrast, 150 nM holo(H-dN) produced morphological phenotypes strikingly different from the other conditions tested. At 15 and 30 min, high levels of mSMC4 signals were detectable on chromatin masses in which entangled thin chromatin fibers were observed (*Figure 3B, D*). At 150 min, although mSMC4 signals on chromatin decreased, a cluster of rod-shaped, mitotic chromosome-like structures with mSMC4-positive axes was clearly discernible in each of the chromatin masses (*Figure 3B, D*). A blow-up image of a representative example is shown in *Figure 3C*. It is important to note that not only the SP/TP sites in the CAP-H N-tail but also those in the CAP-D2 C-tail were barely phosphorylated in I-HSS (*Figure 2—figure supplement 1A, B*).

Finally, holo(H-crCH) and holo(H-N19D) were subjected to the same add-back assay at the concentration of 150 nM (*Figure 3—figure supplement 1A, B*). We found that both mutant complexes behaved similarly to holo(H-dN) although the loading efficiency of holo(H-N19D) was somewhat lower than that of holo(H-crCH). These results demonstrated that, when the CAP-H N-tail is compromised, condensin I gains the ability to assemble mitotic chromosome-like structures even in interphase extracts.

## Deletion of the CAP-H N-tail enhances topological loading onto circular DNA and increases the frequency of loop formation in vitro

We next compared the activities of holo(WT) and holo(H-dN) in two different functional assays without the aid of *Xenopus* egg extracts. In the first setup, we employed the so-called topological loading assay that had been originally developed to assess an ATP-dependent topological entrapment of

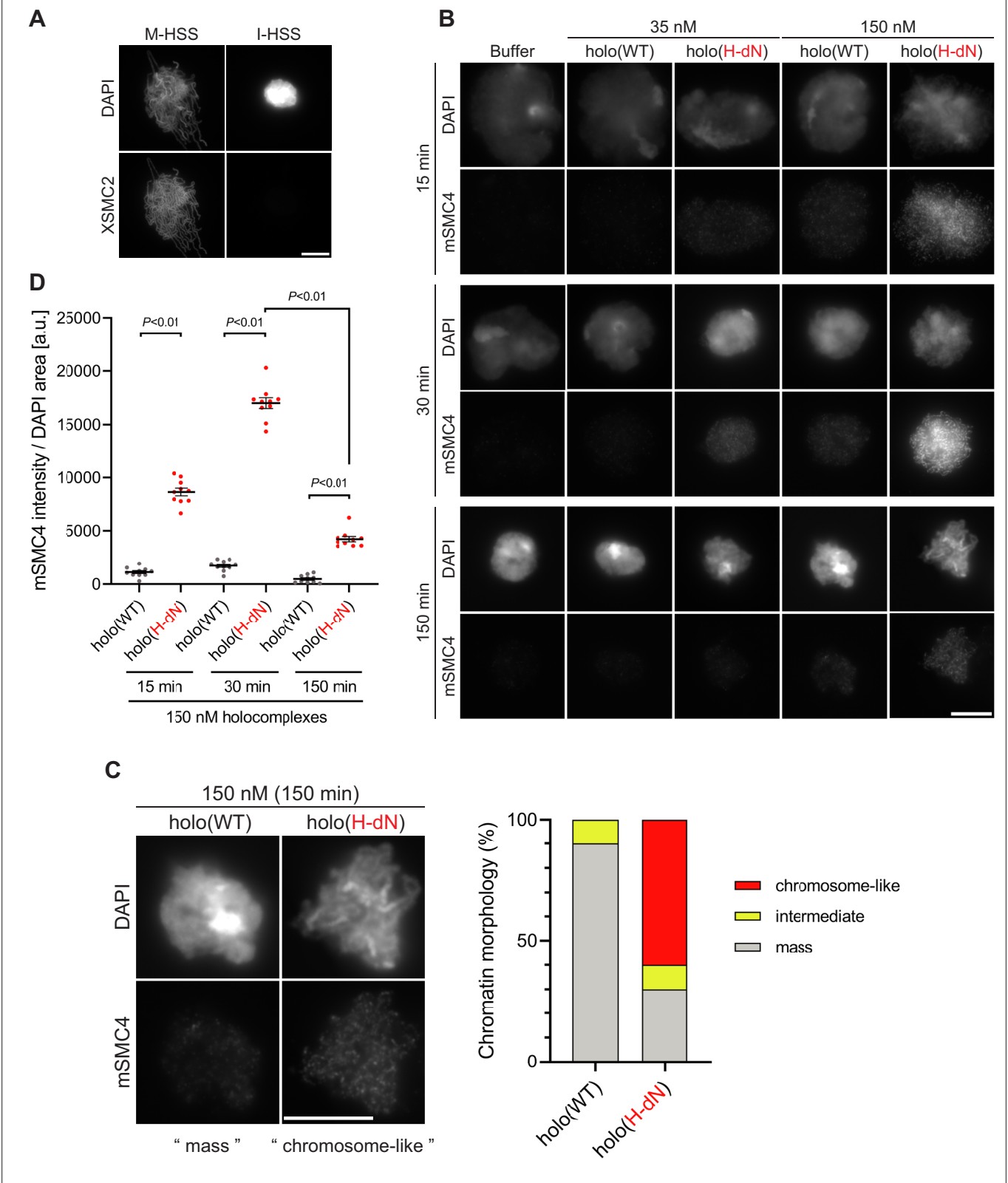

**Figure 3.** Deletion of the CAP-H N-tail enables condensin I to assemble mitotic chromosome-like structures even in interphase extracts. (**A**) Mouse sperm nuclei were incubated with M-HSS (left) and I-HSS (right) for 150 min, and the reaction mixtures were fixed and processed for immunofluorescence labeling with an antibody against XSMC2 (bottom). DNA was counterstained with DAPI (top). Scale bar, 10 μm. (**B**) Mouse sperm nuclei were incubated with condensin-depleted I-HSS that had been supplemented with a control buffer, holo(WT), or holo(H-dN) at a final

*Figure 3 continued on next page*

*Figure 3 continued*

concentration of 35 or 150 nM. After 15, 30, and 150 min, the reaction mixtures were fixed and processed for immunofluorescence labeling with an antibody against mSMC4. DNA was counterstained with DAPI. Shown here is a representative image from over 10 chromosome clusters examined per condition. Scale bar, 10 μm. (**C**) Blow-up images of a chromatin 'mass' assembled with 150 nM holo(WT) at 150 min and a cluster of mitotic 'chromosome-like' structures assembled with 150 nM holo(H-dN) at 150 min in the experiment shown in (B). Frequencies of the chromosomal phenotypes observed under the two conditions, which include 'intermediate' structures, are plotted on the right (*n* = 10 for each condition). Scale bar, 10 μm. (**D**) Quantification of mSMC4 intensities on interphase chromatin when added holo(WT) or holo(H-dN) at 150 nM. The graph shows the intensity of mSMC4 per the DAPI area (*n* = 10 masses of chromatin) shown in (B). The error bars represent the mean ± standard error of the mean (SEM). The p values were assessed by Tukey's multiple comparison test after obtaining a significant difference with two-way analysis of variance (ANOVA). A dataset from a single representative experiment out of three repeats is shown. Another set of the reproduced result was also shown in *Figure 3—figure supplement 1*.

The online version of this article includes the following source data and figure supplement(s) for figure 3:

**Source data 1.** Microsoft excel of non-normalized data corresponding to *Figure 3D*.

**Figure supplement 1.** Deletion of and mutations in the conserved helix enables condensin I to assemble mitotic chromosome-like structures even in interphase extracts.

**Figure supplement 1—source data 1.** Microsoft excel of non-normalized data corresponding to *Figure 3—figure supplement 1B*.

---

nicked circular DNA by the cohesin ring (*Murayama and Uhlmann, 2014*). Briefly, holo(WT) and holo(H-dN) were incubated with nicked circular or linearized DNA in the absence of nucleotides or the presence of ATP or AMP-PNP, and then condensin–DNA complexes were recovered on beads by immunoprecipitation. After being washed with a high-salt buffer, the samples were split into two, and the DNA and proteins that remained on the beads were analyzed by agarose gel electrophoresis (*Figure 4A*) and immunoblotting (*Figure 4B*). Under this condition, virtually no linear DNA was recovered with either holo(WT) or holo(H-dN) regardless of the presence or absence of nucleotides (*Figure 4A*). In contrast, nicked circular DNA was recovered on the beads with holo(WT) in an ATP-dependent manner. Notably, a significantly higher amount (~2.5-fold) of nicked circular DNA was recovered with holo(H-dN) than with holo(WT) (*Figure 4A, C*).

In the second setup, a loop extrusion assay using single DNA molecules was performed (*Sakata et al., 2021*; *Kinoshita et al., 2022*). In brief, holo(WT) and holo(H-dN) were labeled with Alexa Fluor 488 through a HaloTag (*Figure 1—figure supplement 1B*) and subjected to the assay using U-shaped DNA. We found that holo(WT) and holo(H-dN) formed DNA loops on ~17% and ~33% of the U-shaped DNA, respectively, in the presence of ATP (*Figure 4D*). Time-lapse images of DNA loop extrusion events by holo(WT) and holo(H-dN) are shown in *Figure 4E*. Despite the difference in the frequency of loop formation, no significant differences were observed between holo(WT) and holo(H-dN) in other parameters, such as the rate of loop extrusion, the loop duration time, or the loop size formed (*Figure 4F–H*). Neither holo(WT) nor holo(H-dN) supported loop formation in the absence of ATP (*Figure 4—figure supplement 1*). Taken together, these results show that deletion of the CAP-H N-tail affects the initial interactions with DNA, but not the core activity responsible for the expansion of DNA loops.

## Discussion

Previous studies showed that mitotic phosphorylation of Cut3/SMC4 regulates the nuclear import of condensin in fission yeast (*Sutani et al., 1999*) and that phosphorylation of Smc4/SMC4 slows down the dynamic turnover of condensin on mitotic chromosomes in budding yeast (*Robellet et al., 2015*; *Thadani et al., 2018*). In the current study, we have focused on the phosphoregulation of vertebrate condensin I by its kleisin subunit CAP-H.

We have constructed and tested a panel of mutant complexes to provide evidence that the N-tail of the kleisin subunit CAP-H negatively regulates the loading of condensin I and the resultant assembly of mitotic chromosomes in *Xenopus* egg extracts (*Figure 5A*). Recent studies from our laboratory showed that deletion of the CAP-D2 C-tail, which also contains multiple SP/TP sites (*Figure 2—figure supplement 1A*), has little impact on condensin I function as judged by the same and related add-back assays using *Xenopus* egg extracts (*Kinoshita et al., 2022*; *Yoshida et al., 2022*). Thus, the CAP-H N-tail represents the first example of negative regulatory elements that have been identified in vertebrate condensin I. Phosphorylation-deficient mutations (H-N19A) and phosphorylation-mimetic

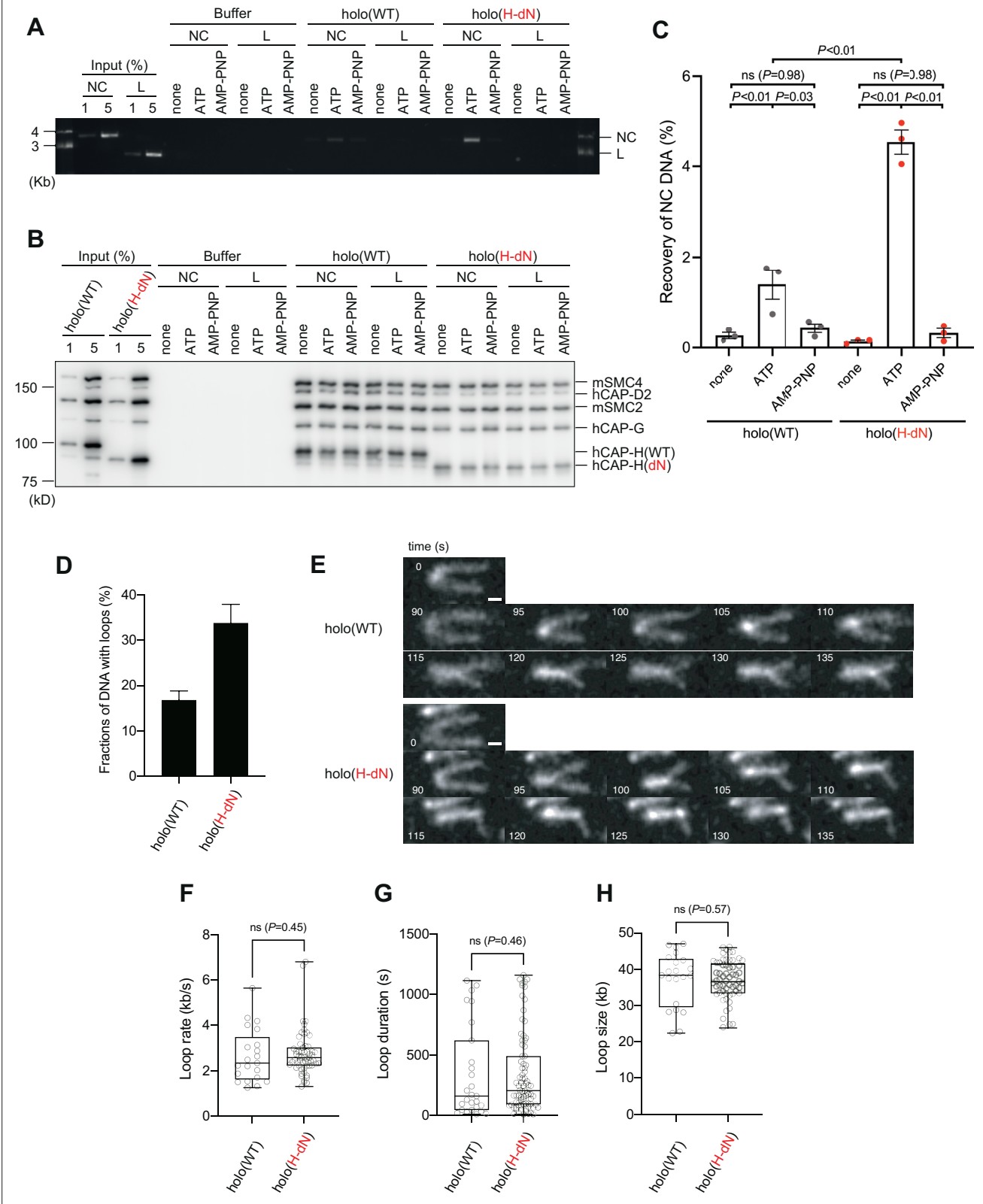

**Figure 4.** Deletion of the CAP-H N-tail enhances topological loading onto circular DNA and increases the frequency of loop formation in vitro. (**A–C**) Topological loading assay. (**A**) Loading reactions onto nicked circular DNA (NC) or linearized DNA (L) were set up with a control buffer, holo(WT) or holo(H-dN) in the absence of nucleotides (none) or the presence of ATP or AMP-PNP. DNAs recovered on beads after a high-salt wash were analyzed by agarose gel electrophoresis. (**B**) Confirmation of the efficiency of immunoprecipitation in the experiment shown in (A). Using the one-tenth volume

*Figure 4 continued on next page*

*Figure 4 continued*

of the recovered samples, immunoblotting analysis was performed with the antibodies indicated. (**C**) Quantification of the recovery of NC DNA in the experiments shown in (A). The error bars represent the mean ± standard error of the mean (SEM) from three independent experiments. The p values were assessed by Tukey's multiple comparison test after obtaining a significant difference with two-way analysis of variance (ANOVA). (**D–H**) Loop extrusion assay. The error bars represent mean ± SEM. The p values shown in (F)–(H) were assessed by a two-tailed Mann–Whitney *U*-test. (**D**) Frequency of DNA loop formation by holo(WT) or holo(H-dN) (*n* = 3, ≥53 DNAs per condition). (**E**) Time-lapse images of DNA loop extrusion events by holo(WT) and holo(H-dN). DNA was stained with Sytox Orange. Scale bar, 1 μm. (**F**) Loop extrusion rate by holo(WT) or holo(H-dN) in the same experiment as in (E) (from three independent experiments, *n* = 21 and 59 for holo(WT) and holo(H-dN), respectively). (**G**) Duration time to maintain DNA loops by holo(WT) or holo(H-dN) in the same experiment as in (E) (from three independent experiments, *n* = 27 and 81 for holo(WT) and holo(H-dN), respectively). (**H**) Loop size produced by holo(WT) or holo(H-dN) in the same experiment as in (E) (from three independent experiments, *n* = 21 and 59 for holo(WT) and holo(H-dN), respectively).

The online version of this article includes the following source data and figure supplement(s) for figure 4:

**Source data 1.** Raw data uncropped gel corresponding to *Figure 4A*.

**Source data 2.** Raw data uncropped blots corresponding to *Figure 4B*.

**Source data 3.** Quantitative data from three independent experiments corresponding to *Figure 4C*.

**Source data 4.** Microsoft excel of quantitative data corresponding to *Figure 4D, F, G and H*.

**Figure supplement 1.** ATP dependency of the loop extrusion activities.

**Figure supplement 1—source data 1.** Microsoft excel of quantitative data corresponding to *Figure 4—figure supplement 1*.

mutations (H-N19D) in the N-tail decelerate and accelerate condensin I loading, respectively, allowing us to propose the following working model (*Figure 5B*). The SMC2-kleisin gate is closed when the CAP-H motif I binds to the SMC2 (*Hassler et al., 2019*) and prevents its untimely opening during interphase. The CH located in the middle of the N-tail could play a critical contribution to this stabilization. Upon mitotic entry, multisite phosphorylation of the N-tail relieves the stabilization, allowing the opening of the DNA entry gate, hence, the loading of condensin I onto chromosomes. Thus, the kleisin N-tail of vertebrate condensin I could act as a 'gatekeeper' of the SMC2-kleisin gate. It is possible that, analogous to the Cdk1 inhibitor Sic1 (*Örd et al., 2019*), multisite phosphorylation of the CAP-H N-tail sets up an ultrasensitive switch-like response that can be activated at a certain threshold level of phosphorylation, thereby ensuring robust cell cycle regulation of condensin I loading. Although our model predicts that the SMC2 neck–kleisin interface is used as a DNA entry gate, we are aware that several studies reported evidence arguing against this possibility (e.g., *Houlard et al., 2021*; *Shaltiel et al., 2022*). Our current data do not exclude other models.

It should be added that CAP-H2, the kleisin subunit of condensin II, lacks the N-terminal extension that corresponds to the CAP-H N-tail. Thus, the negative regulation by the kleisin N-tail reported here is not shared by condensin II. Interestingly, however, a recent study from our laboratory has shown that the deletion of the CAP-D3 C-tail causes accelerated loading of condensin II onto chromatin (*Yoshida et al., 2022*). It is therefore possible that condensins I and II utilize similar IDR-mediated regulatory mechanisms, but they do so in different ways.

In terms of the regulatory role of the CAP-H N-tail, it would be worthy to discuss the model previously proposed by *Tada et al., 2011*. According to their model, aurora B-mediated phosphorylation of the CAP-H N-tail allows its direct interaction with the histone H2A N-tail, which in turn triggers condensin I loading onto chromatin. Accumulating, multiple lines of evidence, however, strongly argue against this model: (1) aurora B is not essential for single chromatid assembly in *Xenopus* egg extracts (*MacCallum et al., 2002*) or in a reconstitution assay (*Shintomi et al., 2015*); (2) the H2A N-tail is dispensable for condensin I-dependent chromatid assembly in the reconstitution assay (*Shintomi et al., 2015*); (3) even whole nucleosomes are not essential for condensin I-mediated assembly of chromatid-like structures (*Shintomi et al., 2017*). The current study demonstrates that the deletion of the CAP-H N-tail accelerates, rather than decelerates, condensin I loading, providing an additional piece of evidence against the model proposed by *Tada et al., 2011*.

Our results show that, when the CAP-H N-tail function is compromised, the mutant condensin I complexes gain the ability to assemble mitotic chromosome-like structures even in interphase extracts. This raises the intriguing possibility that vertebrate condensin I can escape the tight requirement for Cdk1 phosphorylation under certain conditions. Finally, it should be mentioned that, although our recombinant condensin I complexes work excellently in the add-back assays using *Xenopus* egg

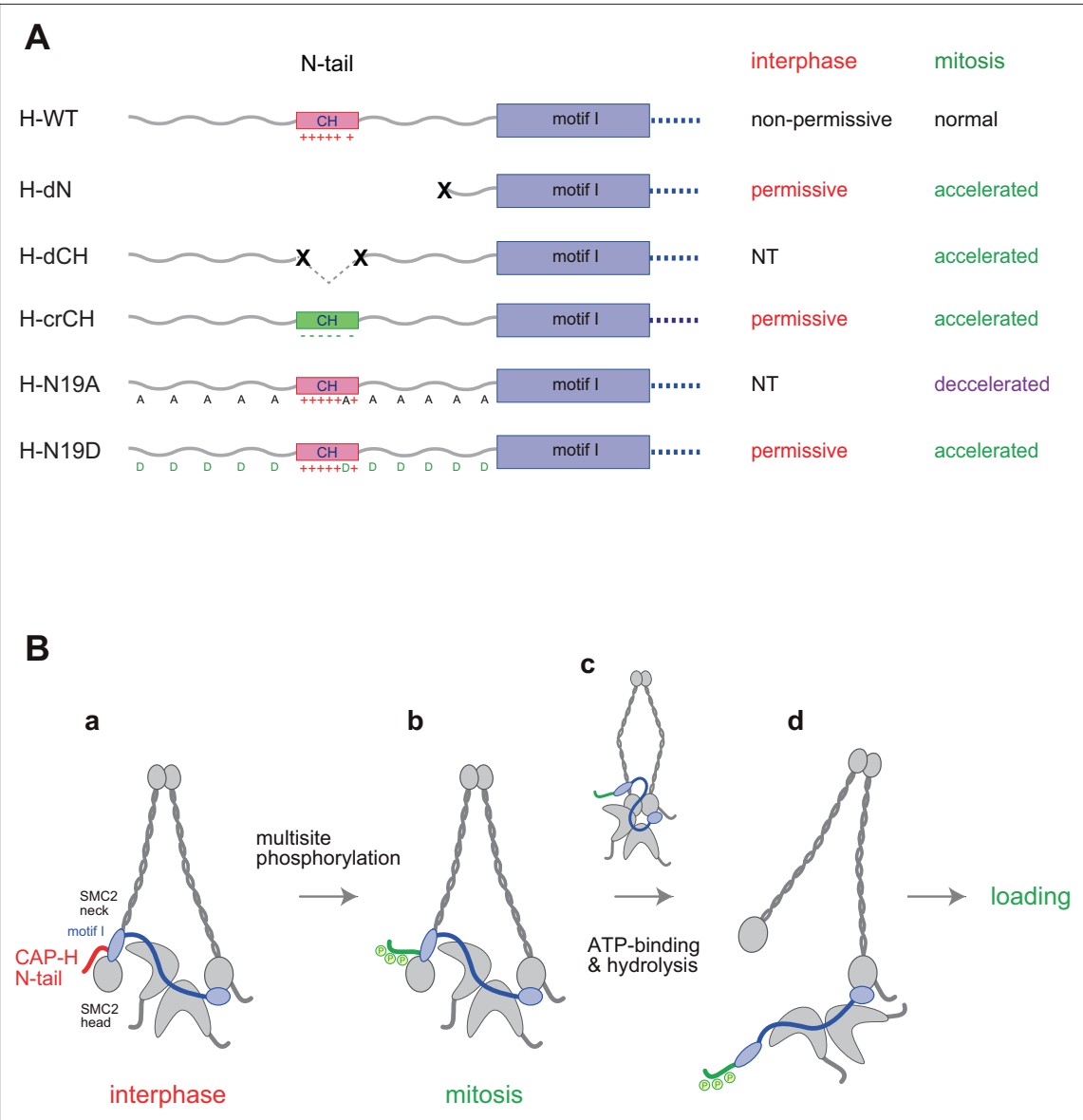

**Figure 5.** Regulation of condensin I by the N-tail of its kleisin subunit. (**A**) Summary of the mutant complexes tested in the current study. NT: not tested. (**B**) Model of cell cycle regulation of condensin I loading. The SMC2-kleisin gate is closed when the CAP-H motif I binds to the SMC2 neck (**a**). In our model, this binding is stabilized by the CAP-H N-tail, thereby preventing its untimely opening during interphase. The conserved helix located in the middle of the N-tail contributes to this stabilization, possibly through direct interaction with the SMC2 head or neck. Upon mitotic entry, multisite phosphorylation of the N-tail relaxes this stabilization (**b**). ATP binding and hydrolysis by the SMC subunits trigger the opening of the DNA entry gate, thereby enabling condensin I to load onto chromosomes (**c, d**). Thus, the kleisin N-tail of vertebrate condensin I could act as a 'gatekeeper' of the SMC2-kleisin gate.

extracts (*Kinoshita et al., 2015*; *Kinoshita et al., 2022*) (the current study), we have so far been unsuccessful in using these recombinant complexes to recapitulate positive DNA supercoiling or chromatid reconstitution, both of which require proper Cdk1 phosphorylation in vitro (*Kimura et al., 1998*; *Shintomi et al., 2015*). Faithfully reconstituting the key activities of condensin I using the recombinant subunits and dissecting multiple phosphorylation events required for its activation are important issues to be addressed in the future.

# Materials and methods

**Key resources table**

| Reagent type (species) or resource | Designation | Source or reference | Identifiers | Additional information |
|---|---|---|---|---|
| Cell line (*Spodoptera frugiperda*) | Sf9 insect cells | Thermo Fisher Scientific | 12659017 | |
| Cell line (*Spodoptera frugiperda*) | Sf9 insect cells | Thermo Fisher Scientific | B825-01 | |
| Cell line (*Trichoplusia ni*) | High Five insect cells | Thermo Fisher Scientific | B85502 | |
| Strain, strain background (*Escherichia coli*) | MAX Efficiency DH10bac Competent Cells | Thermo Fisher Scientific | 10361012 | |
| Biological sample (*Xenopus laevis*) | *Xenopus laevis* eggs | Hamamatsu Seibutsu-Kyozai | RRID:NXR_0031 | Female, adult frogs |
| Biological sample (*Mus musculus*) | *Mus musculus* sperm nuclei | *Mus musculus* cauda epididymis (BALB/c×C57BL/6J)F1; **Shintomi et al., 2017** | N/A | Male, adult mice |
| Antibody | anti-XSMC4 (rabbit polyclonal) | **Hirano and Mitchison, 1994** | In-house: AfR8L | WB (2 µg/ml) |
| Antibody | anti-XSMC2 (rabbit polyclonal) | **Hirano and Mitchison, 1994** | In-house: AfR9-6 | WB (1 µg/ml) |
| Antibody | anti-XSMC2 (rabbit polyclonal, biotin-labeled) | **Hirano and Mitchison, 1994** | In-house: AfR9 | IF (1 µg/ml) |
| Antibody | anti-XCAP-D2 (rabbit polyclonal) | **Hirano et al., 1997** | In-house: AfR16L | WB (1 µg/ml) |
| Antibody | anti-XCAP-G (rabbit polyclonal) | **Hirano et al., 1997** | In-house: AfR11-3L | WB (1 µg/ml) |
| Antibody | anti-XCAP-H (rabbit polyclonal) | **Hirano et al., 1997** | In-house: AfR18 | WB (0.7 µg/ml) |
| Antibody | anti-XCAP-D3 (rabbit polyclonal) | **Ono et al., 2003** | In-house: AfR196-2L | WB (1 µg/ml) |
| Antibody | anti-mSMC4 (rabbit polyclonal) | **Lee et al., 2011** | In-house: AfR326-3L | WB (1 µg/ml); IF (1 µg/ml) |
| Antibody | anti-mSMC2 (rabbit polyclonal) | **Lee et al., 2011** | In-house: AfR329-4L | WB (1 µg/ml) |
| Antibody | anti-hCAP-D2 (rabbit polyclonal) | **Kimura et al., 2001** | In-house: AfR51-3 | WB (1 µg/ml) |
| Antibody | anti-hCAP-G (rabbit polyclonal) | **Kimura et al., 2001** | In-house: AfR55-4 | WB (1 µg/ml); IP (refer to Materials and methods) |
| Antibody | anti-hCAP-H (rabbit polyclonal) | **Kimura et al., 2001** | In-house: AfR57-4 | WB (1 µg/ml) |
| Antibody | anti-XTopo IIa (rabbit antiserum) | **Hirano and Mitchison, 1993** | In-house: αC1-6 | WB (1/2000) |
| Antibody | anti-hCAP-H pS17 (hHP1) (rabbit polyclonal) | This paper; custom ordered from SIGMA Genosys | In-house: AfR464-3P | WB (1 µg/ml) |
| Antibody | anti-hCAP-H pS76 (hHP2) (rabbit polyclonal) | This paper; custom ordered from SIGMA Genosys | In-house: AfR470-3P | WB (1 µg/ml) |

*Continued on next page*

*Continued*

| Reagent type (species) or resource | Designation | Source or reference | Identifiers | Additional information |
|---|---|---|---|---|
| Antibody | anti-hCAP-D2 pT1339 (hDP1) (rabbit polyclonal) | This paper | In-house: AfR173-4P | WB (1 µg/ml) |
| Antibody | anti-hCAP-D2 pT1384 (hDP2) (rabbit polyclonal) | This paper | In-house: AfR175-4P | WB (1 µg/ml) |
| Antibody | anti-hCAP-D2 pT1389 (hDP3) (rabbit polyclonal) | This paper | In-house: AfR177-4P | WB (1 µg/ml) |
| Antibody | Alexa Fluor 568-conjugated anti-rabbit IgG | Thermo Fisher Scientific | A11036 [RRID: AB_10563566] | IF (1/500) |
| Antibody | Horseradish peroxidase-conjugated anti-rabbit IgG | Vector Laboratories | PI-1000 [RRID: AB_2336198] | WB (1/10,000) |
| Peptide, recombinant protein | hHP1 | This paper; custom ordered from SIGMA Genosys | | [C]PHSASpSPSERV |
| Peptide, recombinant protein | hHP2 | This paper; custom ordered from SIGMA Genosys | | [C]PRLLApSPSSRS |
| Peptide, recombinant protein | Precission Protease | Cytiva | 27-0843-01 | Used in purification of recombinant condensin I; *Kinoshita et al., 2022* |
| Peptide, recombinant protein | Benzonase nuclease | Novagen | 71205 | Used in purification of recombinant condensin I; *Kinoshita et al., 2022* |
| Peptide, recombinant protein | $\lambda$ Protein Phosphatase | New England Biolabs | P0753S | |
| Peptide, recombinant protein | Nt.BspQI nicking endonuclease | New England Biolabs | R0644S | |
| Peptide, recombinant protein | EcoRI restriction enzyme | TaKaRa Bio | 1040A | |
| Peptide, recombinant protein | Serotropin (PMSG) | ASKA Pharmaceutical Co, Ltd | | Used in preparation of HSS; *Shintomi and Hirano, 2018* |
| Peptide, recombinant protein | Gonatropin (HCG) | ASKA Pharmaceutical Co, Ltd | | Used in preparation of HSS; *Shintomi and Hirano, 2018* |
| Recombinant DNA reagent | pUC19 | Genbank: L9137 | RRID: Addgene_50005 | |
| Recombinant DNA reagent | $\lambda$ DNA | New England Biolabs | N3011S | |
| Sequence-based reagent | hCAP-H (CH) deletion-F | This paper | Forward primer | GAACGACTTCTCTACCGACTCTCCC |
| Sequence-based reagent | hCAP-H (CH) deletion-R | This paper | Reverse primer | GTAGAGAAGTCGTTCTGAGGGAAGTC |
| Sequence-based reagent | hCAP-H (WT and dN) forward | This paper | Forward primer | GAAGCGCGCGGAATTCGCCA |
| Sequence-based reagent | hCAP-H (WT) reverse | This paper | Reverse primer | GAAGTACAGGTCCTCAGTGGTAGGTT CCAGGTCGCCCTGCCTAACTAA |

*Continued on next page*

*Continued*

| Reagent type (species) or resource | Designation | Source or reference | Identifiers | Additional information |
|---|---|---|---|---|
| Sequence-based reagent | hCAP-H (dN) reverse | This paper | Reverse primer | GAAGTACAGGTCCTCAGTGGTAGGTT CCAGGTCGCCCTGCCTGACTAA |
| Sequence-based reagent | HaloTag forward | This paper | Forward primer | GAGGACCTGTACTTCCAGTCTGACAAC GACATGGCCGAAATCGGAACT |
| Sequence-based reagent | HaloTag reverse | This paper | Reverse primer | AGCGGCCGCGACTAGTTTATC CGCTGATTTCCAGGGTA |
| Commercial assay or kit | PrimeSTAR Mutagenesis Basal Kit | TaKaRa Bio | R046A | |
| Commercial assay or kit | In-Fusion HD Cloning Kit | TaKaRa Bio | 639650 | |
| Commercial assay or kit | Nucleobond PC100 | MACHEREY-NAGEL GmbH & Co KG | 740573.100 | |
| Chemical compound, drug | Cytochalasin D | Sigma-Aldrich | C8273 | Used in preparation of HSS; *Shintomi and Hirano, 2018* |
| Chemical compound, drug | ATP | Sigma-Aldrich | A2383 | |
| Chemical compound, drug | AMP-PNP | Jena Bioscience | NU-407 | |
| Software, algorithm | UNICORN7 | Cytiva | | |
| Software, algorithm | Prism 8 | GraphPad | | |
| Software, algorithm | Olympus cellSens Dimensions | Olympus | | |
| Software, algorithm | Excel | Microsoft | | |
| Software, algorithm | Photoshop | Adobe | | |
| Software, algorithm | ImageJ | https://imagej.nih.gov/ij | | |
| Other | Immobilon Western Chemiluminescent HRP Substrate | Millipore | WBKLS500 | Used detection of immunoblots |
| Other | Alexa Fluor 488-conjugated streptavidin | Thermo Fisher Scientific | S11223 | IF (1/500) |
| Other | Streptavidin | Merck | S4762 | 1 mg/ml (Loop extrusion assay) |
| Other | HaloTag Alexa Fluor 488 Ligand | Promega | G1001 | |
| Other | PD-10 column | Cytiva | 17-0851-01 | |
| Other | Dynabeads Protein A | Thermo Fisher Scientific | 10002D | For immunodepletion with HSS and immunoprecipitation in topological loading assay |
| Other | rProtein A Sepharose Fast Flow | Cytiva | 17-1279-01 | For immunoprecipitation with HSS |
| Other | Glutathione Sepharose 4B | Cytiva | 17075601 | Used in purification of recombinant condensin I; *Kinoshita et al., 2022* |
| Other | HiTrap Q HP 1 ml | Cytiva | 17115301 | Used in purification of recombinant condensin I; *Kinoshita et al., 2022* |

*Continued on next page*

*Continued*

| Reagent type (species) or resource | Designation | Source or reference | Identifiers | Additional information |
|---|---|---|---|---|
| Other | Amicon Ultra-15 | Millipore | UFC905024 | Used in ultrafiltration of purified condensin I; *Kinoshita et al., 2022* |
| Other | DAPI | Roche | 10236276001 | IF (2 µg/ml) |
| Other | GelRed | Biotium | 41003 | |
| Other | Sytox Orange | Thermo Fisher Scientific | S11368 | |

## Preparation of recombinant condensin I complexes

DNA constructs for expressing condensin I subunits in insect cells were described previously (*Kinoshita et al., 2022*). Among them, a special note on the hCAP-H sequence is placed here because multiple cDNA sequences encoding different isoforms of hCAP-H with different lengths have been deposited in the database. Among them, the longest annotated polypeptide of hCAP-H is 741 amino-acid long (NP_056156.2). For historical reasons, our laboratory has been using a sequence of 730 amino-acid long as our reference sequence, whose translation starts at the second methionine of NP_056156.2 (*Onn et al., 2007*; *Kinoshita et al., 2015*; *Kinoshita et al., 2022*). The sequence corresponding to the first 11 amino acids of NP_056156.2 is completely missing in the ortholog of *Xenopus laevis*, and is not conserved in the ortholog of *Mus musculus* (in the latter case, the corresponding sequence is 8 amino-acid long rather than 11 amino-acid long). It is therefore very unlikely that the polypeptide of 730 amino-acid long used in the current study behaves differently from that of 741 amino-acid long. For this reason, the sequence of 730 amino-acid long is used as a 'full-length' hCAP-H in the current study. The recombinant expression vector containing mSMC2 and mSMC4 cDNAs was described previously (*Kinoshita et al., 2015*). The cDNA sequences for hCAP-D2, -G, and -H codon optimized for the cabbage looper *Trichoplusia ni* were synthesized (GeneArt Gene Synthesis Service) and cloned into the pFastBac1 vector (Thermo Fisher Scientific). Deletion of the predicted helix motif in the hCAP-H N-tail was performed using PrimeSTAR Mutagenesis Basal Kit (TaKaRa). Primers used in the deletion were as follows: forward 5′-GAACGACTTCTCTACCGACTCTCCC-3′, reverse 5′-GTAGAGAAGTCGTTCTGAGGGAAGTC-3′. Full-length and N-terminally deleted versions of HaloTag-hCAP-H vectors were constructed by fusing the 3′ ends of the corresponding cDNAs with a HaloTag fragment using In-fusion HD Cloning Kit (TaKaRa). The following primers were used for the construction of these HaloTag-hCAP-H vectors: hCAP-H (WT and dN) forward 5′-GAAGCGCGCGGAATTCGCCA-3′, hCAP-H (WT) reverse 5′-GAAGTACAGGTCCTCAGTGGTAGGTTCCAGGTCGCCCTGCCTAACTAA-3′, hCAP-H (dN) reverse 5′-GAAGTACAGGTCCTCAGTGGTAGGTTCCAGGTCGCCCTGCCTGACTAA-3′, HaloTag forward 5′-GAGGACCTGTACTTCCAGTCTGACAACGACATGGCCGAAATCGGAACT-3′ HaloTag reverse 5′-AGCGGCCGCGACTAGTTTATCCGCTGATTTCCAGGGTA-3′.

Note that the HaloTag was fused to the C-terminus of CAP-H in the current study because we wanted to investigate the effect of the N-terminal deletion of CAP-H. We used N-terminally HaloTagged CAP-H constructs in our previous study (*Kinoshita et al., 2022*). DH10Bac (Thermo Fisher Scientific) was transformed with the plasmid vectors to produce bacmid DNAs, with which Sf9 cells were transfected to generate the corresponding baculoviruses. To produce recombinant condensin I complexes, High Five cells (Thermo Fisher Scientific) were infected with the baculoviruses carrying the condensin I subunits, and the resultant complexes were purified from cell lysates by two-step column chromatography as described previously (*Kinoshita et al., 2022*).

## Antibodies

Primary antibodies used in the present study were as follows: anti-XSMC4 (in-house identifier AfR8L, affinity-purified rabbit antibody), anti-XSMC2 (AfR9-6 for immunodepletion and immunoblotting, biotin-labeled AfR9 for immunofluorescence, affinity-purified rabbit antibody), anti-XCAP-D2 (AfR16L, affinity-purified rabbit antibody), anti-XCAP-G (AfR11-3L, affinity-purified rabbit antibody), anti-XCAP-H (AfR18, affinity-purified rabbit antibody; *Hirano and Mitchison, 1994*; *Hirano et al., 1997*), anti-XCAP-D3 (AfR196-2L, affinity-purified rabbit antibody; *Ono et al., 2003*), anti-mSMC4 (AfR326-3L, affinity-purified rabbit antibody), anti-mSMC2 (AfR329-4L, affinity-purified rabbit

antibody; *Lee et al., 2011*), anti-hCAP-D2 (AfR51-3, affinity-purified rabbit antibody), anti-hCAP-G (AfR55-4, affinity-purified rabbit antibody), anti-hCAP-H (AfR57-4, affinity-purified rabbit antibody; *Kimura et al., 2001*), and anti-XTopo IIα (αC1-6, rabbit antiserum; *Hirano and Mitchison, 1993*). Two custom phospho-specific antibodies against hCAP-H (AfR464-3P and AfR470-3P, affinity-purified rabbit antibodies) were raised against the phosphopeptides hHP1 ([C]PHSASpSPSERV) and hHP2 ([C] PRLLApSPSSRS), respectively (SIGMA Genosys), where cysteines (C) were added for chemical conjugation. The phosphoserines in hHP1 and hHP2 corresponded to pS17 and pS76 in the 730 amino-acid long isoform, respectively, and to pS28 and pS87 in the 741 amino-acid long isoform, respectively (see above for the explanation of the isotypes with different lengths). The epitopes recognized by anti-hHP1 and anti-hHP2 were detectable on holo(WT), but not on holo(H-N19A), that had been preincubated with Δcond M-HSS (*Figure 2—figure supplement 1B*). Each epitope was sensitive to phosphatase treatment (*Figure 2—figure supplement 1C*), and competed with its own phosphopeptide but not with others (*Figure 2—figure supplement 1D*). Three phospho-specific antibodies against hCAP-D2 (AfR173-4P, AfR175-4P, and AfR177-4P, affinity-purified rabbit antibodies) were raised against the phosphopeptides hDP1 ([C]DNDFVpTPEPRR), hDP2 ([C]MTEDEpTPKKTT), and hDP3 ([C]TPKKTpT-PILRA), respectively, and purified according to the procedure described previously (*Kimura et al., 1998*). The phosphothreonines in hDP1, hDP2, and hDP3 corresponded to pT1339, pT1384, and pT1389, respectively (*Figure 2—figure supplement 1A*). Secondary antibodies used in the present study were as follows: Alexa Fluor 568-conjugated anti-rabbit IgG (A11036 [RRID: AB_10563566], Thermo Fisher Scientific), Alexa Fluor 488-conjugated streptavidin (S11223, Thermo Fisher Scientific), and horseradish peroxidase-conjugated anti-rabbit IgG (PI-1000 [RRID: AB_2336198], Vector Laboratories).

## Animals

Female *X. laevis* frogs (RRID:NXR_0031, Hamamatsu Seibutsu-Kyozai) were used to lay eggs to harvest *Xenopus* egg extract (*Hirano et al., 1997*). Frogs were used in compliance with the institutional regulations of the RIKEN Wako Campus. Mice (BALB/c × C57BL/6JF1) for sperm nuclei (*Shintomi et al., 2017*) were used in compliance with protocols approved by the Animal Care and Use Committee of the University of Tokyo (for M. Ohsugi who provided mouse sperm).

## Preparation of *Xenopus* egg extracts

The high-speed supernatant of metaphase-arrested *Xenopus* egg extracts (M-HSS) was prepared as described previously (*Shintomi and Hirano, 2018*). For the preparation of interphase HSS (I-HSS), a low-speed supernatant (LSS) of metaphase-arrested extracts was supplemented with $CaCl_2$ and cycloheximide at final concentrations of 0.4 mM and 100 μg/ml, respectively (*Losada et al., 2002*). After incubation at 22°C for 30 min, the interphase LSS was further fractionated by centrifugation at 50,000 rpm at 4°C for 90 min to yield I-HSS.

## Immunodepletion, add-back assay and immunofluorescence

Immunodepletion of endogenous *Xenopus* condensins I and II from the extracts was performed using Dynabeads Protein A (10002D, Thermo Fisher Scientific) as described previously (Δcond) (*Kinoshita et al., 2022*). The efficiency of immunodepletion was checked every time by immunoblotting. An example of immunodepletion from M-HSS is shown in *Figure 1—figure supplement 1C*. We also confirmed that a similar efficiency of immunodepletion was achieved from I-HSS. Add-back assays using mouse sperm nuclei were performed as described previously (*Kinoshita et al., 2022*). After incubation at 22°C, aliquots of the reaction mixtures were taken at indicated time points and fixed by adding 10 volumes of KMH (20 mM HEPES-KOH [pH 7.7], 100 mM KCl, and 2.5 mM $MgCl_2$) containing 4% formaldehyde and 0.1% Triton X-100 at 22°C for 15 min. The fixed chromatin was sedimented onto a coverslip through 5 ml cushion of XBE2 (10 mM HEPES-KOH [pH 7.7], 100 mM KCl, 2 mM $MgCl_2$, 0.1 mM $CaCl_2$, 5 mM EGTA, and 50 mM sucrose) containing 30% glycerol by centrifugation at 5000 rpm for 10 min. For immunofluorescence, the coverslips were blocked with TBS-Tx containing 1% bovine serum albumin (BSA) for 30 min and then incubated with primary antibodies for 60 min. After being washed three times with TBS-Tx, the coverslips were incubated with secondary antibodies for 60 min. After DNA was stained with DAPI, they were mounted with VectaShield mounting medium (H-1000, Vector Laboratories) onto slides and then observed under an Olympus BX63 fluorescent

microscope equipped with a UPlanSApo ×100/1.40 NA oil immersion lens and an ORCA-Flash 4.0 digital complementary metal oxide semiconductor camera C11440 (Hamamatsu Photonics). The fluorescent images were acquired using CellSens Dimension software (Olympus).

## Quantitative analyses of immunofluorescent images and statistics

Immunofluorescent images acquired by microscopic observation were quantified using ImageJ software (https://imagej.nih.gov/ij). Line profiles of mitotic chromosomes were scanned according to the procedure described previously (*Kinoshita et al., 2022*). For quantification of recombinant condensin I complexes on chromatin, the DAPI- and mSMC4-positive signals from the images were segmented using the threshold function and the integrated densities and the areas were measured. The mSMC4 intensities were divided by the DAPI-positive areas to assess the accumulation of condensin complexes on chromosome clusters. The total number of chromosome cluster images counted for statistical analysis in each experiment is described in the appropriate figure legends. The data were handled with Excel software (Microsoft), and the graphs were drawn using GraphPad Prism software (version 8, GraphPad Software). Tukey's multiple comparison test after obtaining a significant difference with one-way analysis of variance (ANOVA) (*Figure 2B, E*, and *Figure 1—figure supplement 2C*) or two-way ANOVA (*Figures 1C and 3D*, and *Figure 3—figure supplement 1B*).

## Immunoblotting

Denatured protein samples were subjected to sodium dodecyl sulfate–polyacrylamide gel electrophoresis (SDS–PAGE) and transferred onto a nitrocellulose membrane (Cytiva). The membrane was blocked with 5% skim milk in TBS-Tw (20 mM Tris–HCl [pH 7.5], 150 mM NaCl, and 0.05% Tween 20) for 30 min and incubated with primary antibodies diluted with 1% BSA in TBS-Tw for 1 hr. After being washed three times with TBS-Tw, the membrane was incubated with a peroxidase-conjugated secondary antibody diluted with TBS-Tw for 1 hr. After being washed three times with TBS-Tw, the membrane was treated with a chemiluminescence substrate for peroxidase (WBKLS500, Merck). The chemiluminescent image was acquired using an image analyzer (Amersham Imager 680, Cytiva).

## Detection of phosphoepitopes in *Xenopus* egg extracts

For immunoprecipitation, rProtein A Sepharose (rPAS, Cytiva) was equilibrated with TBS. 5 µl of rPAS were coupled with 5 µg anti-hCAP-G antibody at 4°C for 2 hr with rotation and then washed three times with KMH. Holo(WT) or holo(H-N19A) was added to Δcond M-HSS or I-HSS at a final concentration of 35 nM, along with 1/50 vol of 50× energy mix (50 mM Mg-ATP, 500 mM phosphocreatine, and 2.5 mg/ml creatine kinase [pH 7.5]) and incubated at 22°C for 150 min. The final reaction volume was 15 µl. The reaction mixtures were immunoprecipitated with the anti-hCAP-G antibody-coupled rPAS on ice for 30 min. The beads were washed with 150 µl KMH containing 20 mM β-glycerophosphate and 0.1% Triton X-100, and then centrifuged (7000 rpm) at 4°C for 1 min. The recovered beads were further washed three times with 200 µl of KMH containing 20 mM β-glycerophosphate and 0.1% Triton X-100, followed by two repeated washes with 200 µl of KMH containing 20 mM β-glycerophosphate. The immunoprecipitants were analyzed by immunoblotting. For dephosphorylation, the immunoprecipitants bound to the beads were washed with 200 µl of KMH containing 20 mM β-glycerophosphate and 0.1% Triton X-100, and then washed twice with 200 µl of KMH. The washed beads were incubated with 10 µl of dephosphorylation buffer (1× NEB buffer for PMP supplemented with 1 mM MnCl$_2$) with or without 100U $\lambda$ Protein Phosphatase (P0753S, $\lambda$ PPase) (New England Biolabs). A cocktail of phosphatase inhibitors (50 mM NaF, 10 mM Na$_3$VO$_4$, and 50 mM EDTA at final concentrations) was added to one of the reaction mixtures containing $\lambda$ PPase. The reaction mixtures were incubated at 30°C for 30 min and analyzed by immunoblotting. For the peptide competition assay, a primary antibody solution containing AfR464-3P or AfR470-3P (20 µg/ml) was preincubated with or without phosphopeptides (hHP1 or hHP2 at a final concentration of 100 µg/ml) at room temperature for 60 min in 50 µl of TBS. The mixtures were then diluted 20 times with TBS-Tw containing 1% BSA and used for probing immunoblots.

## Topological loading assay

A topological loading assay was performed according to the procedure described by *Murayama and Uhlmann, 2014* with minor modifications. Nicked circular DNA and linearized DNA were used as

DNA substrates in this assay. To prepare the nicked circular DNA, negatively supercoiled plasmid DNA (pUC19) purified from *Escherichia coli* cells using Nucleobond PC100 (MACHERREY-NAGEL GmbH & Co KG) was treated with the nicking enzyme Nt.BspQI (R0644S, New England Biolabs). The linearized DNA was prepared by digesting the negatively supercoiled DNA with *Eco* RI (1040A, TaKaRa Bio). The treated DNAs were extracted with phenol/chloroform/isoamyl alcohol (25:24:1) and then purified by ethanol precipitation. For condensin I loading assay, 2 pmol of holo(WT) or holo(H-dN) and 90 ng of nicked or linearized DNA were mixed in reaction buffer (35 mM Tris–HCl [pH 7.5], 1 mM Tris-(2-carboxyethyl)phosphine (TCEP), 50 mM NaCl, 7 mM $MgCl_2$, 0.003% Tween 20, and 15% glycerol) and then supplemented with ATP or AMP-PNP (NU-407, Jena Bioscience) at a final concentration of 2 mM. The final reaction volume was 15 µl. The mixture was incubated at 32°C for 1 hr. Note that the final concentration of NaCl in the loading reaction was 77 mM because of carry-over from the storage buffer of purified condensin I. The loading reaction was terminated by adding 85 µl of IP buffer (35 mM Tris–HCl [pH 7.5], 0.5 mM TCEP, 500 mM NaCl, 10 mM EDTA, 5% glycerol, and 0.35% Triton X-100). Condensin I–DNA complexes were then immunoprecipitated with 10 µl of Dynabeads protein A (10002D, Thermo Fisher Scientific) that had been pre-coupled with 2.5 µg of affinity-purified anti-hCAP-G. The beads were washed three times with 750 µl of Wash Buffer-I (35 mM Tris–HCl [pH 7.5], 0.5 mM TCEP, 750 mM NaCl, 10 mM EDTA, and 0.35% Triton X-100), and then once with 500 µl of Wash Buffer-II (35 mM Tris–HCl [pH 7.5], 0.5 mM TCEP, 100 mM NaCl, and 0.1% Triton X-100). To check the efficiency of immunoprecipitation, protein samples were retrieved from a one-tenth volume of the beads and analyzed by immunoblotting. The remaining beads were treated with 15 µl of deproteinization solution (10 mM Tris–HCl [pH 7.5], 1 mM EDTA, 50 mM NaCl, 0.75% SDS, and 1 mg/ml proteinase K) and incubated at 50°C for 20 min. The recovered DNAs were electrophoresed on a 1% agarose gel in Tris-acetate EDTA (TAE) buffer, and stained with GelRed (Biotium). The fluorescent images were acquired using an image analyzer (Amersham Imager 680, Cytiva). The signal intensities of nicked circular DNAs recovered by holo(WT) and holo(H-dN) were quantified using ImageJ software. Three independent experiments were performed and assessed by Tukey's multiple comparison test after obtaining a significant difference with two-way ANOVA.

### Loop extrusion assay

The HaloTag holocomplexes were labeled with HaloTag Alexa Fluor 488 Ligand (Promega) at room temperature for 30 min. The labeled complexes were applied onto a PD-10 column (Cytiva) to remove unbound fluorescent ligands. A loop extrusion assay was performed as described previously (*Sakata et al., 2021*; *Kinoshita et al., 2022*). In brief, coverslips were coated with 1 mg/ml streptavidin in ELB++ (10 mM HEPES–KOH [pH 7.7], 50 mM KCl, 2.5 mM $MgCl_2$, and 1 mg/ml BSA) for 30 min. After assembling microfluidic flow cells that can switch the flow direction, they were incubated with ELB++ for 30 min, and then washed with T20 buffer (40 mM Tris–HCl [pH 7.5], 20 mM NaCl, and 0.2 mM EDTA). Biotin-labeled $\lambda$ DNA in T20 buffer was attached to the coverslips for 5 min, and then unbound DNA was washed off with imaging buffer (50 mM Tris–HCl [pH 7.5], 50 mM NaCl, 5 mM $MgCl_2$, 3 mM ATP, and 1 mM DTT). The flow direction was switched, and the flow cells were washed with the same buffer for another 10 min. Then, 1 nM condensin in the imaging buffer was introduced into the flow cells at 20 µl/min for 1.5 min, and washed off with the same buffer at 20 µl/min. For visualization of Sytox Orange (22 nM)-stained DNA and Alexa 488-labeled condensin, 488 and 561 nm lasers were used, respectively. Images were taken every 5 s for 20 min after introducing condensin. The images were analyzed using the ImageJ software. To determine the loop size, the fluorescence intensity of looped DNA was divided by that of the entire DNA molecule for each image, and multiplied by the length of the entire DNA molecule (48.5 kb). The loop rate was obtained by averaging the increase in looped DNA size per second. The loop duration was calculated by measuring the time from the start of DNA loop formation until the DNA loop became unidentifiable. The data in *Figure 4F-H* were statistically analyzed by a two-tailed Mann–Whitney *U*-test.

### Materials availability

All unique/stable reagents generated in this study are available from the corresponding author.

## Acknowledgements

We thank F Inoue, H Watanabe, and M Ohsugi for their help with mouse sperm nuclei preparation, and members of the Hirano lab for critically reading the manuscript. This work was supported by Grant-in-Aid for Scientific Research/KAKENHI (#17K15070 [to ST], #19H05755 and #22H02551 [to KS], #19K06499 [to KK], #20K15723 [to MMY], #20H05937 [to TN], and #18H05276 and #20H05938 [to TH]) and by JST PRESTO (JPMJPRK4 [to TN]).

## Additional information

### Funding

| Funder | Grant reference number | Author |
| --- | --- | --- |
| Japan Society for the Promotion of Science | #17K15070 | Shoji Tane |
| Japan Society for the Promotion of Science | #19H05755 | Keishi Shintomi |
| Japan Society for the Promotion of Science | #22H02551 | Keishi Shintomi |
| Japan Society for the Promotion of Science | #19K06499 | Kazuhisa Kinoshita |
| Japan Society for the Promotion of Science | #20H05937 | Tomoko Nishiyama |
| Japan Society for the Promotion of Science | #18H05276 | Tatsuya Hirano |
| Japan Society for the Promotion of Science | #20H05938 | Tatsuya Hirano |
| Precursory Research for Embryonic Science and Technology | JPMJPRK4 | Tomoko Nishiyama |
| Japan Society for the Promotion of Science | #20K15723 | Makoto M Yoshida |

The funders had no role in study design, data collection, and interpretation, or the decision to submit the work for publication.

### Author contributions

Shoji Tane, Conceptualization, Data curation, Formal analysis, Funding acquisition, Writing - original draft, Writing - review and editing; Keishi Shintomi, Kazuhisa Kinoshita, Resources, Supervision, Funding acquisition, Writing - review and editing; Yuko Tsubota, Data curation, Writing - review and editing; Makoto M Yoshida, Data curation, Funding acquisition, Writing - review and editing; Tomoko Nishiyama, Supervision, Funding acquisition, Writing - review and editing; Tatsuya Hirano, Conceptualization, Supervision, Funding acquisition, Validation, Writing - original draft, Project administration, Writing - review and editing

### Author ORCIDs

Shoji Tane http://orcid.org/0000-0002-0209-347X
Keishi Shintomi http://orcid.org/0000-0003-0484-9901
Kazuhisa Kinoshita http://orcid.org/0000-0002-0882-4296
Yuko Tsubota http://orcid.org/0000-0002-7479-2747
Makoto M Yoshida http://orcid.org/0000-0002-0618-1717
Tomoko Nishiyama http://orcid.org/0000-0002-8349-6536
Tatsuya Hirano http://orcid.org/0000-0002-4219-6473

### Ethics

Female Xenopus laevis frogs (RRID: NXR_0031, Hamamatsu Seibutsu-Kyozai) were used to lay eggs to harvest Xenopus egg extract (Hirano et al., 1997). Frogs were used in compliance with the institutional regulations of the RIKEN Wako Campus. Mice (BALB/c × C57BL/6J F1) for sperm nuclei (Shintomi et

al., 2017) were used in compliance with protocols approved by the Animal Care and Use Committee of the University of Tokyo (for M. Ohsugi who provided mouse sperm).

## Decision letter and Author response
Decision letter https://doi.org/10.7554/eLife.84694.sa1
Author response https://doi.org/10.7554/eLife.84694.sa2

## Additional files

### Supplementary files
• MDAR checklist

### Data availability
All data generated or analyzed during this study are included in the manuscript and supporting files.

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
