## [Editor Report]

This important study provides insight into how vertebrate condensin I activity is restricted to mitosis. Using in vitro experiments in *Xenopus* extracts, convincing evidence is presented that phosphorylation of the N-terminal extension of CAP-H relieves an inhibitory activity that prevents condensin I loading onto chromosomes. The authors present a speculative model for the mechanistic basis of this inhibition which will provide inspiration for future investigations in the chromosome condensation field.

---

## [Decision Letter]

[Editors' note: this paper was reviewed by Review Commons.]

---

## [Author Response]

Reviewer #1 (Evidence, reproducibility and clarity (Required)):CommentsThe work described in this manuscript starts with an in-silico analysis of the primary amino-acid sequence of CAP-H proteins that reveals the presence in vertebrate orthologs of an N-terminal extension of ~80 amino acids in length which contains 19 serine or threonine residues and also, in its centre, a stretch of conserved basic amino acids predicted to form a helix. These features suggest a regulatory module. Using *Xenopus* egg extracts depleted of endogenous condensins and supplemented with recombinant condensin I holocomplexes, either wildtype or mutants, the authors show that deleting the N-terminal tail of CAP-H, or just the central helix (CH), increases the association condensin I with chromatin in mitotic egg extracts and accelerates the formation of mitotic chromosomes. Interestingly, they also show that deleting the N-tail enables a substantial amount of condensin I to associate with chromatin in interphase extracts and to form chromosome-like structures, while WT condensin I cannot. Using in vitro assays and naked DNA as substrate, the authors further show that removing the N-terminal tail of CAP-H improves both the topological (salt-resistant) association of condensin I with DNA and it loop extrusion activity. These experiments appear to me as are clear and robust. They convincingly reveal that N-tail of human CAP-H hinders the binding of condensin I to DNA and both its loop-extrusion and chromosome-shaping activities. However, the mechanism through which such hindrance is achieved remains elusive (see major comments 1-3).A complementary part of the work tackles the important question of the cell cycle control of such counteracting effect. Using newly-designed antibodies against two phospho-serine residues within the tail, the authors provide evidence that the tail is phosphorylated in mitosis-specific manner. This points towards phosphorylation as a biological mean to modulate the effect of the tail on condensin's binding during the cell cycle. In support to this idea, using non-phosphorylatable or phosphomimic substitutions of all the serine and threonine residues within the tail (n=19), including one substitution within the CH domain (Ser 70), the authors show that nonphosphorylatable mutations (H-N19A) or phosphomimic mutations (H-N19D) respectively reduce or improve condensin I binding to chromatin in mitotic egg extracts. This suggests that the phosphorylation of the N-terminal tail in mitosis might relieve its negative effect on condensin I binding to chromatin. The weaknesses I see in this part of the study concern (1) the validation of the phospho-antibodies, which appears to me as insufficiently described (major comment 4), (2) the possibility the bulk changes in amino acids (n=19 out of 80) could impact the folding of the central helix (minor comment X) and (3) that some substitutions could impact the binding of condensin I by different mechanisms (minor comment X).Major comments:1. On the model. The authors propose that the N-tail could stabilise an interaction between the N-terminal part of CAP-H and SMC2's neck, which would restrain the transient opening of a DNA entry gate within the ring, necessary for the topological engagement of DNA and loop formation. Although the model is sound, I see no direct data that support it in the manuscript. Such model predicts that a CAP-H protein containing or not the N-terminal tail (or the central helix) should exhibit different binding strengths to SMC2 in vitro. It seems to me that the authors could easily test this prediction using the recombinant proteins they produced in the context of this study.

We thank the reviewer for pointing out this important issue. To test whether the CAP-H N-tail indeed contributes to the stabilization of the SMC2-kleisin gate, we set up a highly sophisticated functional assay described by Hassler et al. (2019). The authors used this assay to demonstrate that an N-terminal fragment of kleisin (engineered to be cleaved by TEV protease) is released from the rest of the condensin complex in an ATPdependent (i.e., head-head engagement-dependent) manner. We reasoned that this assay is most powerful to prove our hypothesis in a mechanistically relevant context. We envisioned that the CAP-H fragment lacking its N-tail can readily be released whereas the CAP-H fragment retaining its N-tail is more difficult to be released (because of the postulated stabilization of the SMC2-CAP-H interaction). Despite substantial efforts in making TEV-cleavable constructs and in testing various releasing conditions, we have not been able to recapitulate the ATP-dependent release even with the holo(H-dN) construct. Thus, unfortunately, this trial enabled us to neither prove nor disprove our hypothesis.

We are fully aware that the full reconstitution of ATP-dependent and phosphorylation-stimulated gate-opening reaction in vitro is a very important direction in the future. It is beyond the scope of the current study, however.

2. On ATP-hydrolysis. Given the importance of ATP hydrolysis for the engagement of condensin into a topological mode of association with DNA and for its loop extrusion activity, I suggest that the authors measure the impact of the N-tail and of the CH domain on the rate of ATP hydrolysis by condensin I holocomplexes. I suppose that it can be relatively easily done (PMID: 9288743) using the recombinant WT and mutant versions they purified in the course of this study.

We appreciate this constructive comment. In fact, we did a preliminary experiment and found that ATPase activities (either in the absence or presence of DNA) were not significantly different between holo(WT) and holo(H-dN). We were not surprised with this result because our previous study on condensin II indicated that enhanced ATP hydrolysis by a class of mutant complexes is not directly coupled to their enhanced association with chromosomes (Yoshida et al., 2022, *eLife*). We consider that other functional assays, such as the topological loading assay and the loop extrusion assay shown in the current manuscript, are more informative assays to address ATP-dependent activities of the condensin complexes.

3. A conundrum with previous work? In Kimura et al. Science 1998 (PMID: 9774278), the lab of Tatsuya Hirano has shown that *Xenopus* condensin I purified from mitotic egg extracts induces the supercoiling of plasmid DNA in vitro, but fails to do so when it is purified from interphase egg extracts. This echoes the inhibitory effect of the N-tail of the topological binding of condensin I described in the current manuscript. However, using a gel shift assay, Kimura et al. 1998 also provide evidence that interphase and mitotic condensin I bind plasmid DNA in vitro with similar efficiencies. At first sight, this prior observation seems to contradict the idea that the N-tail of CAP-H restrains DNA binding unless it is phosphorylated in mitosis. Is it possible that the in vitro binding assays used in Kimura et al. 1998 and in this work might assess different modes of binding? I suggest that this apparent conundrum should to be discussed.

We thank the reviewer for following our early studies. As discussed below, we are confident that our conclusion reported in the current study by no means contradicts our previous observations.

We reason that the confusion expressed by the reviewer stems from intrinsic, technical limitations of the gel-shift assay. Such limitations become apparent especially when it is applied to the functional analyses of complicated protein machines such as condensins. For instance, the DNA-binding activity of condensin I detected by the gel-shift assay is neither ATP-dependent nor phosphorylation-dependent (Kimura and Hirano, 1997; Kimura et al., 1998). It is fundamentally different from the ATP-dependent activities detected by the topological loading and loop extrusion assays reported in the current study (It remains unknown whether the two activities are stimulated by mitotic phosphorylation). Thus, the DNA-binding activity detected by the gel-shift assay does not reflect “productive” DNA interactions that depend on ATP hydrolysis in vitro. We therefore consider that gel-shift analyses of holo(WT) and holo(H-dN) would not produce any useful information.

Related to that, could it be possible for the authors to assess the impact of the N-tail on the salt-sensitive binding of condensin to DNA, i.e. by reproducing the topological binding assay but omitting the high salt washes? I guess this information could be useful to fully apprehend the impact of the N-tail on the binding of condensin.

When we set up the topological loading assay, we actually tested a low-salt wash condition that the reviewer suggests here. Although a much higher level of DNA recovery was observed with the low-salt condition than with the high-salt wash condition, no nucleotide dependency was detectable with the low-salt condition. Moreover, no difference in DNA recovery between holo(WT) and holo(H-dN) was observed. Thus, the low-salt condition allowed us to detect the “bulk” DNA-binding activity that is equivalent to that detected by the gel-shift assay. These results were fully consistent with the discussion above.

4. Validation of phospho-antibodies and by extension showing the phosphorylation of the tail. The newly-designed phospho-serine antibodies used in this study to show that the N-tail is phosphorylated at serine 17 and serine 76 in mitosis (Figure 2-figure supplement 1) are, in my view, not characterized enough. This piece of data is key to substantiate the idea that the tail is phosphorylated in mitosis. Yet, showing that these antibodies detect epitopes on WT condensin I from mitotic egg extracts but not on the H-N19A counterpart, nor on WT condensin I from interphase extracts, does not demonstrate the phospho-specificity of such antibodies. I suggest that a PPase treatment should be conducted to assess the phospho-specificity of these antibodies. Moreover, since the lab of Tatsuya Hirano has the know-how to deplete Cdc2/CDK1 from *Xenopus* egg extract, such strategy could/should be used to further validate the antibodies and assess to which extent the Ntail is phosphorylated in a Cdc2-dependent manner.

We have performed two sets of experiments to confirm the specificity of the phosphoepitopes recognized by anti-hHP1 and anti-hHP2. In the first set, we performed a phosphatase treatment assay. Holo(WT) that had been preincubated with Δcond M-HSS was immunoprecipitated using an antibody against hCAP-G, treated with λ protein phosphatase in the presence or absence of phosphatase inhibitors, and analyzed by immunoblotting using anti-hHP1 and anti-hHP2. The results (now shown in Figure 2-figure supplement 1C) demonstrated that the epitopes recognized by anti-hHP1 and anti-hHP2 are sensitive to phosphatase treatment. In the second set, we performed a phosphopeptide competition assay. Holo(WT) that had been preincubated with Δcond M-HSS was immunoprecipitated and subjected to immunoblotting. The membranes were triplicated and probed with anti-hHP1 in the presence of no competing peptide, hHP1 or hHP2. Similarly, another set of triplicated membranes was probed with anti-hHP2 in the presence of no competing peptide, hHP1 or hHP2. We found that the signal recognized by anti-hHP1 competed with hHP1, but not with hHP2, and that the signal recognized by anti-hHP2 competed with hHP2, but not with hHP1. The results (now shown in Figure 2-figure supplement 1D) demonstrated the sequence specificity of the phosphoepitopes recognized by the two antibodies. The procedures for these experiments have been described in Materials and methods.

Because Cdk1 depletion from M-HSS creates an HSS equivalent to I-HSS, we do not consider that the suggested experiment will provide additional information.

Minor comments:1. The impact of the 19 mutations, A or D, introduced within the tail on the folding of the central helix? The idea that the negative effect of the N-tail is relieved by phosphorylation is based on the chromatin binding phenotypes exhibited by the H-N19D or H-N19A mutant holocomplexes, in which 19 amino-acids out of 80 have been modified, include one in the central helix. The authors also provide evidence that the central helix (CH) located within the tail plays a key role in the negative regulation of condensin I binding. Thus, I wonder to which extent the folding of the central helix could be impacted by the mutations introduced in the tail and notably the one within the central helix itself. Could the author assess the structure of mutated tails using Alphofold and/or discuss this point?

According to the reviewer’s suggestion, we performed structure predictions using Alphafold2, and found that neither the N19A nor N19D mutations alter the original prediction of helix formation that was made for the wild-type CH sequence. A conventional secondary structure prediction using Jpred4 reached the same conclusion.

2. Phosphorylation of serine 70 in the central helix by Aurora-B kinase? A prior study by Tada et al. (PMID: 21633354) has shown (1) that serine 70 of the N-tail of hCAP-H is phosphorylated by Aurora-B kinase, (2) that the mutation S70A reduces the binding of condensin I to chromatin in HeLa cells and (3) that hCAP-H interacts with histone H2A in an Aurora-B dependent manner. This draws a picture in which the phosphorylation of Ser70 by Aurora-B would improve condensin I binding to chromatin by promoting an interaction between hCAP-H and histone H2A/nucleosomes.Intriguingly, Ser 70 in Tada et al. correspond to the serine residue located within the conserved central helix analysed in this study, and this Ser70 residue is mutated in the H-N19D or H-N19A holocomplexes that show reduced chromatin binding in this study. This raises the question as what could be the contribution of the S70A or S70D substitution to the chromatin binding phenotypes shown by the H-N19D or H-N19A holocomplexes. This is not discussed in the manuscript, and the authors do not cite this earlier work (PMID: 21633354) in their manuscript. Is there any reason for that? I suggest it should be cited and discussed.

We thank the reviewer for bringing up this issue. In many respects, we do not trust the data reported by Tada et al. (2011) and the resultant model they proposed. Previous and emerging lines of evidence reported from our own and other laboratories indicate that histones compete with condensins for DNA binding, strongly arguing against the possibility that histone H2A acts as a “chromatin receptor” for condensins. We formally discussed and criticized the Tada 2011 model in our previous publications, which included Shintomi et al. (2015) NCB, Shintomi et al. (2017) Science, Hirano (2016) Cell and Kinoshita/Hirano (2017) COCB. We thought that those were enough. That said, we also consider that the reviewer is right. The current study demonstrates that the deletion of the CAP-H N-tail accelerates, rather than decelerates, condensin I loading, providing an additional line of evidence that argues against the Tada model. A critical comparison between the Tada 2011 model and our current model would benefit the readers. In the revised manuscript, we have added the following discussion:

“In terms of the regulatory role of the CAP-H N-tail, it would be worthy to discuss the model previously proposed by Tada et al. (2011). According to their model, aurora B-mediated phosphorylation of the CAP-H N-tail allows its direct interaction with the histone H2A N-tail, which in turn triggers condensin I loading onto chromatin. Accumulating lines of evidence, however, strongly argue against this model: (i) aurora B is not essential for single chromatid assembly in *Xenopus* egg extracts (MacCallum et al., 2002) or in a reconstitution assay (Shintomi et al., 2015); (ii) the H2A N-tail is dispensable for condensin I-dependent chromatid assembly in the reconstitution assay (Shintomi et al., 2015); (iii) even whole nucleosomes are not essential for condensin I-mediated assembly of chromatid-like structures (Shintomi et al., 2017). The current study demonstrates that the deletion of the CAP-H N-tail accelerates, rather than decelerates, condensin I loading, providing an additional piece of evidence against the model proposed by Tada et al. (2011).”

3. Other minor comments– Please provide a microscope image of DNA loop in Figure 4D.

In the revised manuscript, we have provided a set of time-lapse images of loop extrusion events catalyzed by holo(WT) and holo(H-dN) in Figure 4E.

– The authors do not compare the kleisin of condensin I with the one of condensin II with respect to the features tackled in this work. Given the different behaviours condensin I and II, such comparison could be informative for the readers.

We thank the reviewer for this constructive comment. In the revised manuscript, we have added the following statement:

“It should also be added that CAP-H2, the kleisin subunit of condensin II, lacks the N-terminal extension that corresponds to the CAP-H N-tail. Thus, the negative regulation by the kleisin N-tail reported here is not shared by condensin II.”

– The authors do not reference the work of Robellet et al. Genes & Dev (2015) (PMID: 25691469) on the regulation of condensin binding in budding yeast by an SMC4 phospho-tail. I suggest that the analogy should be discussed.

According to the reviewer’s comment, we have added the following statements at the beginning of Discussion.

“Previous studies showed that mitotic phosphorylation of Cut3/SMC4 regulates the nuclear import of condensin in fission yeast (Sutani et al. 1999) and that phosphorylation of Smc4/SMC4 slows down the dynamic turnover of condensin on mitotic chromosomes in budding yeast (Robellet et al. 2015; Thadani et al. 2018). In the current study, we have focused on the phosphoregulation of vertebrate condensin I by its kleisin subunit CAP-H.”

– In the introduction section, lane 5, the sentence "Many if not all eukaryotic species have two different condensin complexes" appears inappropriate since budding and fission yeast cells possess a single condensin complexes, similar to condensin I in term of primary amino-acid sequence.

We thought that the original wording “Many if not all” was good enough to imply that some species, which include budding yeast and fission yeast, have only a single condensin complex. To make it clear, however, we have modified the sentence in the revised manuscript as follows:

“Many eukaryotic species have two different condensin complexes although some species including fungi have only condensin I.”

– page 4; typo: motif I and V bind to the SMC neck and the SMC4 cap regions, respectively. Should read SMC2 neck.

The reviewer is right. It should read the SMC2 neck. Corrected.

– Are the data and the methods presented in such a way that they can be reproduced?Yes– Are the experiments adequately replicated and statistical analysis adequate?Yes– Are prior studies referenced appropriately? Not all of them (see above) – Are the text and figures clear and accurate?YesCross-consultation commentsI consider the comments from all reviewers as helpful for the authors.Reviewer #1 (Significance (Required)):SummaryCondensins are genome organisers of the family of SMC ATPase complexes and are best characterized as the drivers of mitotic chromosome assembly (condensation). It is acknowledged that condensins shape mitotic chromosomes by massively associating with DNA upon mitotic entry (loading step) and by folding chromatin fibres into arrays of loops, most likely through an ATP-dependent extrusion of DNA into loops, as seen in vitro. What remains unclear, however, are the mechanisms by which condensins load onto DNA and fold it into arrays of loops in vivo, and how these reactions are coupled with the cell cycle, i.e. restricted mostly to mitosis. Condensins are ring shaped pentamers that change conformation upon ATP-hydrolysis. in vitro studies suggest that condensins bind DNA via ATP-hydrolysis-independent, direct electrostatic contacts between condensin subunits and DNA. Such electrostatic contacts are salt-sensitive in in-vitro assays. Upon ATP-hydrolysis, condensins engage into an additional mode of binding that is resistant to high salt concentration and likely to be topological in nature. Both modes of association are necessary to form DNA loops. Vertebrates possess two types of condensin complexes, condensin I and II, each composed of a same SMC2-SMC4 ATPase core but associated with two different sets of three non-SMC subunits; a kleisin and two HEAT-repeat proteins. Condensin II is nuclear during interphase and stably binds chromatin upon mitotic entry, while condensin I is located in the cytoplasm during interphase and binds chromatin in a dynamic manner upon nuclear envelope breakdown. How the spatiotemporal control of condensin I and II is achieved remains poorly understood. Previous studies have shown that the phosphorylation of condensin I by mitotic kinases, such as CDK1, Aurora-B and Polo, play a positive role in its binding to chromatin and/or its functioning, but the underlying mechanisms remain to be characterised. In this manuscript, Shoji Tane and colleagues provide good evidence that the N-terminal tail of the human kleisin subunit of condensin I, hCAP-H, serves as a regulatory module for the cell-cycle control of condensin I binding to chromatin and chromosome shaping activity. The authors clearly show that the N-tail of CAP-H hinders the binding of condensin I to chromatin in *Xenopus* egg extracts and, using in vitro assays, that the N-tail also hinders the topological association of condensin I with DNA and its loop extrusion activity. The authors provide additional data suggesting that the phosphorylation of the N-tail of CAP-H, in mitosis, relieves its inhibitory effect on condensin I binding. Based on their findings, Tane et al. propose a model suggesting that the N-terminal tail of CAP-H constitutes a gate keeper that maintains condensinrings in a closed conformation that is unfavourable for topological binding to DNA, and whose locking effect is relieved in mitosis by phosphorylation.Taken as a whole, this work has the potential to reveal a molecular basis for the cell cycle regulation of condensin I in vertebrate cells and as such to significantly improve our understanding of the integrated functioning condensin I. The characterisation of the inhibitory effect of the N-tail on condensin binding to chromatin and to naked DNA in vitro is well described, the data are clear and robust and the results convincing. On the other hand, some of the data on the phospho-regulation appear to me as are more debatable and I think that some of the results described here deserve to be discussed in the context of previous works. Finally, I see no data in the manuscript that directly supports the mechanistic model proposed by the authors, while it seems to me that such model could have been easily tested exprimentally. Thus, I suggest that Tane and colleagues should perform a couple of relatively easy experiments to strengthen their claims and that a few omitted prior studies on the topic should be referenced and discussed.Reviewer #2 (Evidence, reproducibility and clarity (Required)):The manuscript reveals that the N-terminal region of CAPH could play a role in regulating condensin I activity, using a range of in vitro methods. They propose that the N-terminal region of CAPH inhibits complex activity, and this is turned off upon deletion or phosphorylation, by using truncations, phospho-mimics or phospho-deficient mutations.While the results are interesting to the field, and helps to address the question as to how condensin complexes are controlled in a cell cycle dependent manner, some key data and controls are necessary to ensure the conclusion is robust.Main commentsWhat is meant by "unperturbed I-HSS" on page 7, ie membrane containing versus membrane free or condensin depleted?

We apologize for having created unnecessary confusion. We meant that the “unperturbed I-HSS” is the “undepleted I-HSS”. As far as the issue of membrane-containing vs membrane-free is concerned, we explicitly mentioned that “we used membrane-free I-HSS in the following experiments” several lines above. In the revised manuscript, we have revised the statement accordingly.

In many of the protein gels eg figure 4B, the bands for SMC2 and 4 are more intense that the non-SMC components. The method for protein purification also does not include a size exclusion step to ensure sample homogeneity. Authors should perform some sort of quality control checks on samples such as analytical gel filtration or mass photometry to ensure stoichiometry/homogeneity. This is particularly important for samples eg the N19A, where activity is reduced compared to wild-type as poor protein behaviour could create false negative results.

As the reviewer is fully aware, the reconstitution and purification of multiprotein complexes, such as condensins, is by no means an easy task. We notice that many groups in the field share common concerns about sample homogeneity and subunit stoichiometry, and that these concerns cannot completely be eliminated even after size exclusion chromatography. Because the current study handles a large number of mutant complexes, we consider that the purification by two-step column chromatography is the most practical approach. We do not notice any abnormal behaviors of holo(H-N19A) in the processes of expression and purification. It is also important to emphasize that the H-N19D mutations cause the completely opposite phenotype. Taken all together, we are confident of our current conclusions.

That said, in the revised manuscript, we have added the following statement in Results:

“Although we cannot rule out the possibility that the introduction of multiple mutations into the N-tail causes unforeseeable adverse effects on protein conformations, these results supported the idea that ….”

Loop extrusion assays in figure 4D-G shows no example data i.e. no pictures or videos of loops being formed. These should also be included.

In the revised manuscript, we have provided a set of time-lapse images of loop extrusion events catalyzed by holo(WT) and holo(H-dN) in Figure 4E.

Given the mutant holo(H-dN) has higher activity than wild-type, a negative control such as holo(H-dN) without ATP or holo(H-dN) ATPase deficient mutant should also be measured in loop extrusion assays, to ensure the activity is derived from the ATPase activity.

In the revised manuscript, we have added loop formation data for both holo(WT) and holo(H-dN) in the absence or presence of ATP (Figure 4-figure supplement 1). We are confident that both complexes support loop extrusion strictly in an ATP-dependent manner.

According to the methods, this work performs the same loop extrusion assay as described in Kinoshita et al., 2022, however, in Kinoshita et al., wild type condensin I makes loops in 30-50% of DNA molecules, where in this study the percentage is less than half that. Can the author please explain the discrepancy given the same method was used?

First of all, we wish to remind the reviewer that the holo(WT) constructs used in the two studies are not identical: CAP-H was N-terminally HaloTagged in all constructs used in Kinoshita et al. (2022), whereas the same subunit was C-terminally HaloTagged in the pair of constructs used in the current study. Because we wanted to compare the activities between the full-length CAP-H and N-terminally deleted version of CAP-H (H-dN), we reasoned that it would be inappropriate to put the HaloTag to the N-terminus of CAP-H. The difference in the constructs could explain the observed discrepancy, even if it might not be the sole reason.

The design of the constructs was accurately described in each manuscript, but the statements were not very explicit about the positions of the HaloTag. To clarify this issue, we have added the following sentences in the revised manuscript:

“Note that the HaloTag was fused to the C-terminus of CAP-H in the current study because we wanted to investigate the effect of the N-terminal deletion of CAP-H. We used N-terminally HaloTagged CAP-H constructs in our previous study (Kinoshita et al., 2022).”

In the concluding statement the author suggests "Upon mitotic entry, multisite phosphorylation of the N-tail relieves the stabilization, allowing the opening of the DNA entry gate, hence, the loading of condensin I onto chromosomes." This seems unlikely as fusion the N-terminus of the of the kleisin to the C-terminus of SMC2 is able to function for yeast (Shaltiel et al. 2022) and condensin II (Houlard et al. 2021), and equivalently in cohesin (Davidson et al. 2019).

We appreciate the reviewer’s concern. In our view, however, the issue of the “DNA entry gate” remains under debate in the SMC field (e.g., Higashi et al. [2020] Mol Cell; Taschner and Gruber [2022] bioRxiv). For instance, Shaltiel et al. (2022) demonstrated that neck-gate fusion constructs can support in vitro activities including topological loading under certain conditions, but also showed that such constructs greatly reduce the cell viability, leaving the possibility that the gate opening is required for some physiological functions.

That said, it is true that the data reported in the current manuscript do not exclude the possibility that the SMC2 neck-kleisin interface is not used as a DNA entry gate for condensin I loading. In the revised manuscript, we have added the following statement in Discussion:

"Although our model predicts that the SMC2 neck-kleisin interface is used as a DNA-entry gate, we are aware that several studies reported evidence arguing against this possibility (e.g., Houlard et al. [2021]; Shaltiel et al. [2022]). Our current data do not exclude other models."

Reviewer #2 (Significance (Required)):This is an interesting story that reveals new insights about condensin regulation.Reviewer #3 (Evidence, reproducibility and clarity (Required)):This paper reveals a role of an N-terminal extension of CAP-H in the regulation of condensin-I activity in *Xenopus* extracts using biochemical reconstitution experiments. The authors demonstrate that a motif in the N-terminal tail that is conserved in vertebrates acts as an inhibitor of condensin I activity. Using several mutant constructs, it is shown that the inhibition by this motif is in turn counteracted by the phosphorylation of neighbouring serine and threonine residues in mitosis, presumably at least in part by Cdk. Mutants that have lost this inhibition are able to condense chromatin into chromatid-like structures more efficiently and to some degree even in interphase extracts. Moreover, one such mutant is characterized in detail by biochemical and biophysical experiments and shown to have increased capacity in salt-stable DNA loading and in DNA loop extrusion.Major comments:This is a beautiful and thorough study that is presented in a clear and concise manner.The main conclusions are well justified. No additional experiments are needed to support them. Replication and statistical analysis appear adequate. The final model is however largely speculative. Recent work has indicated that loading of yeast condensin does not require gate opening. The authors may thus want to include alternative scenarios or remain more vague.

This comment is related to the last comment of Reviewer#2. See above for our response.

The H-N19A mutant has a loss of function phenotype (possibly due to folding problem caused by 19 point mutations rather than lack of phosphorylation), the authors could consider to rescue the phenotype by also including the CH motif mutations in this construct (or make an explanatory statement in the text).

We understand the reviewer’s logic here, but overlaying additional mutations into the H-N19A mutations could cause an unforeseeable effect, potentially making the interpretation of the outcome complicated.

We also wish to point out that it may be inappropriate to regard the phenotype exhibited by holo(H-N19A) as a simple loss-of-function phenotype. This is because the opposite, accelerated loading phenotype exhibited by holo(H-dN) can be regarded as a consequence of loss of negative regulation. Like holo(H-dN), the phosphomimetic mutant complex holo(H-N19D) displayed an accelerated loading phenotype, fully supporting our conclusion. In the revised manuscript, we have added the following statement in Results:

“Although we cannot rule out the possibility that the introduction of multiple mutations into the N-tail causes unforeseeable adverse effects on protein conformations, these results supported the idea that ….”

Albeit not necessary for the main conclusions, the authors could possibly significantly strengthen their study by testing for binding partners of the N-tail and the CH motif by running AlphaFold predictions against the condensin I subunits.

We appreciate this constructive comment. We attempted to predict possible interactions between SMC2 and a CAP-H fragment containing its N-tail and motif I using ColabFold (Mirdita et al., 2022, Nat. Methods). The algorism excellently predicted the proper folding of the CAP-H motif I and its interaction with the SMC2 neck. Under this condition of predictions, however, the N-tail remained largely disordered (except for the CH), and no interaction with any part of SMC2 was predicted. The same was true when the N19D mutations were introduced in the N-tail sequence. Thus, this trial did not provide much information about the potential interaction target(s) of the CAP-H N-tail.

The efficiency of depletion of condensin subunits from I-HSS extracts is not documented (in contrast to M-HSS extracts – Figure 1-figure supplement 1C). While any condensin remaining in these extracts might not be active (or interfering), the authors may want to at least comment on this in the text.

We check the efficiency of immunodepletion every time by immunoblotting and confirm that a high level of depletion is achieved from both M-HSS and I-HSS. According to the reviewer’s comment, the following statement was placed in Materials and methods:

“The efficiency of immunodepletion was checked every time by immunoblotting. An example of immunodepletion from M-HSS was shown in Figure 1-figure supplement 1C. We also confirmed that a similar efficiency of immunodepletion was achieved from I-HSS.”

The authors should include information on the quantification of chromatid morphology. Is the analysis based on chromatids taken from the same images/imaging session, from technical replicates, biological replicates?

In the revised manuscript, we have added statements on image presentation and experimental repeats in the appropriate figure legends and methods section. During the revision process, we repeated the experiments shown in Figure 1-figure supplement 2, and obtained the same results. In the revised manuscript, the original set of data has been replaced with the new set of data along with panel C (Quantification of the intensity of mSMC4 per DNA area).

Minor comment:The colour scheme in Figure 5A is confusing. Use less colour? The orange and red colours are moreover quite similar.

According to the reviewer’s comment, we have modified Figure 5A.

Reviewer #3 (Significance (Required)):The findings provide new insights into how condensin-I activity is restricted outside of mitosis. It was previously assumed that this regulation was (largely) due to the exclusion of condensin I from the nucleus prior to nuclear envelope breakdown. This study shows that another pathway is contributing to the regulation and implies that controlling condensin I activity is more important than previously appreciated. Whether all residual nuclear condensin I is inactivated, remains to be determined. The physiological impact of loss of autoinhibition on chromosome segregation and cell cycle progression also remains to be uncovered. The observed effects are robust and appear significant. Future research on condensin I using reconstitution will likely benefit from being able to control or eliminate the self-inhibition.This reviewer has expertise on the biochemistry and structural biology of SMC protein complexes.Reviewer #4 (Evidence, reproducibility and clarity (Required)):Mitotic chromosome formation is a cell cycle-regulated process that is crucial for eukaryotic genome stability. The chromosomal condensin complex promotes chromosome condensation, but the temporal control over condensin function is only scantly understood. In this impressive manuscript, "Cell cycle-specific loading of condensin I is regulated by the N-terminal tail of its kleisin subunit", Tane and colleagues provide important new insight into the cell cycle-regulation of condensin. The authors identify a kleisin N-tail that acts as a negative regulator of condensin's DNA interactions. Removal of this N-tail, or its cell cycle-dependent phosphorylation, relieves inhibition and activates condensin. This is a simple, yet very important story, that advances our molecular understanding of chromosome formation. The experiments are performed to a very high technical standard and support the authors conclusions. This manuscript is highly suitable for publication in any molecular biology journal, once the authors have considered the following points.1. Introduction. (a) The authors could better explain their own prior work (Kimura et al. 1998), which has identified the condensin XCAP-D2 and XCAP-H as the targets of phosphoregulation. The current manuscript explains the role of XCAP-H phosphorylation.

According to the reviewer’s comment, we have added the following sentence in Introductions:

“The major targets of mitotic phosphorylation identified in these studies included the CAP-D2 and CAP-H subunits.”

b) Given the limited knowledge about condensin cell cycle regulation, it seems prudent to provide a brief summary of what is known. Fission yeast Smc4 phosphorylation regulates condensin nuclear import (Sutani et al. 1999), while budding yeast Smc4 phosphorylation slows down the dynamic turnover of the condensin complex on chromosomes (Robellet et al. 2015 and Thadani et al. 2018).

We appreciate this constructive comment. According to the reviewer’s comment, we have added the following statements at the beginning of Discussion.

“Previous studies showed that mitotic phosphorylation of Cut3/SMC4 regulates the nuclear import of condensin in fission yeast (Sutani et al. 1999) and that phosphorylation of Smc4/SMC4 slows down the dynamic turnover of condensin on mitotic chromosomes in budding yeast (Robellet et al. 2015 and Thadani et al. 2018). In the current study, we have focused on the phosphoregulation of vertebrate condensin I by its kleisin subunit CAP-H.”

2. Extracts were mixed with mouse sperm nuclei. If there is a reason why mouse rather than *Xenopus* sperm nuclei were used, this would be interesting to know.

The original motivation for introducing mouse sperm nuclei into *Xenopus* egg extracts was to test the functional contribution of nucleosomes to mitotic chromosome assembly. When mouse sperm nuclei are incubated with an extract depleted of the histone chaperone Asf1, the assembly of octasomes can be suppressed almost completely. Remarkably, we found that even under this “nucleosome-depleted” condition, mitotic chromosome-like structures can be assembled in a manner dependent on condensins (Shintomi et al., 2017, Science). *Xenopus* sperm nuclei cannot be used in this type of experiment because they endogenously retain histones H3 and H4 and are therefore competent in assembling octasomes even in the Asf1-depleted extract. During this study, we realized that the use of mouse sperm nuclei in *Xenopus* egg extracts provides additional and deep insights into the basic mechanisms of mitotic chromosome assembly. For instance, the functional contribution of condensin II to chromosome assembly could be observed more prominently when mouse sperm nuclei are used as a substrate than when *Xenopus* sperm nuclei are used (Shintomi et al., 2017, Science). We suspected that the slow kinetics of nucleosome assembly on the mouse sperm substrate creates an environment in favor of condensin II’s action. For these reasons, our laboratory now extensively uses mouse sperm nuclei for the functional analyses of condensin II (Yoshida et al., 2022. *eLife*) and other purposes (Kinoshita et al., 2022, JCB). Yoshida et al. (2022) used experimental approaches analogous to the current study, and found that the deletion of the CAP-D3 C-tail, causes accelerated loading of condensin II. One of the long-term goals in our laboratory is to critically compare and contrast the actions of condensin I and condensin II in mitotic chromosome assembly. Thus, the use of the same substrate in the two complementary studies can be fully justified.

During the preparation of this response, we realized that the readers would benefit from a brief statement about the comparison between condensin I and condensin II. In the revised manuscript, we have added the following statement in Discussion:

“It should also be added that CAP-H2, the kleisin subunit of condensin II, lacks the N-terminal extension that corresponds to the CAP-H N-tail. Thus, the negative regulation by the kleisin N-tail reported here is not shared by condensin II. Interestingly, however, a recent study from our laboratory has shown that the deletion of the CAP-D3 C-tail causes accelerated loading of condensin II onto chromatin (Yoshida et al., 2022). It is therefore possible that condensins I and II utilize similar IDR-mediated regulatory mechanisms, but they do so in different ways.”

3. Page 5. "we next focused on the conserved helix (CH) […], that is enriched with basic amino acids." Based on the provided sequence alignment, the helix contains an equal number of both basic and acidic residues. Is it correct to characterize this helix as positively charged?

The reviewer is right. In the revised manuscript, we have used a more neutral expression as follows:

“…we next focused on the conserved helix (CH) […], that contains conserved basic amino acids.”

4. To prevent N-tail phosphorylation, the authors create a (H-N19A) allele, referring to Cdk promiscuity. Cdk cooperation with other mitotic kinases can also be expected. Nevertheless, in case the authors created a variant with only the 4 Cdk consensus sites mutated, it would be interesting to know its consequences.

We consider that this is a reasonable question. In our early experiments, we noticed that introduction of multiple SP/TP sites in the non-SMC subunits of condensin I including CAP-H caused a relatively mild phenotype in mitotic chromosome assembly in *Xenopus* egg extracts. Then we found that the deletion of the CAP-H N-tail caused a very clear, accelerated loading phenotype, prompting us to focus on the regulatory function of the CAP-H N-tail. As the reviewer correctly points out, the current study does not pinpoint the number and position of target sites involved in the proposed phosphoregulation by the CAP-H N-tail. We wish to address this important issue in the near future, along with reconstitution of the phosphoregulation using purified components.

5. Figure 2-figure supplement 1A, a second region of mitotic condensin phosphorylation is XCAP-D2. The authors state that XCAP-D2 phosphorylation does not impact on condensin function in their assays. This is very relevant to the current paper, so it would be good to see the Yoshida et al. 2022 ELife publication (in press) as an accompanying manuscript.

We thank the reviewer for pointing out this issue, but it is not necessarily clear to us what the reviewer requests. In the original manuscript, we cited Yoshida et al. (2022) in Discussion as follows:

“Recent studies from our laboratory showed that the deletion of the CAP-D2 C-tail, which also contains multiple SP/TP sites (Figure 2-figure supplement 1A), has little impact on condensin I function as judged by the same and related add-back assays using *Xenopus* egg extracts (Kinoshita et al., 2022; Yoshida et al., 2022).”

We believe that the current statement is good enough.

6. One of the authors' most striking results is chromosome formation in interphase egg extracts using condensin (H-dN). At the same time, condensin (H-dN) is unable to support DNA supercoiling or chromosome reconstitution with recombinant components. More emphasis might be placed on this important piece of information, and possible reasons should be discussed. Can Cdk-treatment restore condensin (H-dN) biochemical activity? If not, then condensin (H-dN) might have lost more than just an inhibitory Ntail. The cohesin N-tail is thought to fulfil a positive role during DNA loading (Higashi et al. 2020). Could it be that the condensin N-tail encompasses both positive and negative roles?

We were also surprised to find that holo(H-dN) gains the ability to assemble mitotic chromosome-like structures in interphase extracts. It should be stressed, however, that the formation of mitotic chromosome-like structures in I-HSS requires a much higher concentration (150 nM) than the standard concentration used in M-HSS (35 nM). Thus, the deletion of the CAP-H N-tail alone cannot make the condensin I complex fully active in I-HSS. We think that the negative regulation by the CAP-H N-tail is not the sole mechanism responsible for the very tight cell cycle regulation of condensin I function. We emphasize this important point by mentioning that “our results uncover one of the multi-layered mechanisms that ensure cell cycle-specific loading of condensin I onto chromosomes” in Summary.

At the end of Discussion, we describe the limitations of the current study: “we have so far been unsuccessful in using these recombinant complexes to recapitulate positive DNA supercoiling or chromatid reconstitution, both of which require proper Cdk1 phosphorylation in vitro”. We are fully aware that full reconstitution of phosphorylation dependent activation of condensin I in vitro is one of the most important directions in the future.

Although we currently do not have any evidence to suggest that the H N-tail has a positive role, we do not exclude such a possibility.

7. Here comes my main question for the authors (though I should stress that I do not expect an answer for publication in a Review Commons journal). The authors now have a unique opportunity to gain key mechanistic insight into condensin by answering the question, 'how does the kleisin N-tail inhibit condensin'? The authors allude to a model in which the N-tail interacts with Smc2 to close/obstruct the kleisin N-gate, through which the DNA likely enters the condensin ring. Can the authors biochemically recapitulate an interaction between an isolated N-tail (or N-terminal section of XCAPH) and Smc2? Does Cdk phosphorylation alter this interaction?

This comment is related to Comment #1 of Reviewer#1. See above for our response.

Minor points.8. The condensin loop extrusion results would benefit from a supplementary movie or time-series, to illustrate the comparison. Details of how loop rate, duration and sizes were assessed should be added to the methods section.

In the revised manuscript, we have provided a set of time-lapse images of loop extrusion events catalyzed by holo(WT) and holo(H-dN) in Figure 4E. We have also added the following explanations for how the parameters of loop extrusion reactions were assessed in Materials and methods:

“To determine the loop size, the fluorescence intensity of the looped DNA was divided by that of the entire DNA molecule for each image, and multiplied by the length of the entire DNA molecule (48.5 kb). The loop rate was obtained by averaging the increase in looped DNA size per second. The loop duration was calculated by measuring the time from the start of DNA loop formation until the DNA loop became unidentifiable.”

9. Figure 2-figure supplement 1A legend, "hHP4" should probably read "hHP2".

The reviewer is right. It should read hHP2. Corrected.

Reviewer #4 (Significance (Required)):See above.